# PPDiff: Diffusing in Hybrid Sequence-Structure Space for Protein-Protein Complex Design

**Zhenqiao Song** [*1]  **Tianxiao Li** [2]  **Lei Li** [1]  **Martin Renqiang Min** [2]

## Abstract

Designing protein-binding proteins with high affinity is critical in biomedical research and biotechnology. Despite recent advancements targeting specific proteins, the ability to create high-affinity binders for arbitrary protein targets on demand, without extensive rounds of wet-lab testing, remains a significant challenge. Here, we introduce PPDiff, a diffusion model to jointly design the sequence and structure of binders for arbitrary protein targets in a non-autoregressive manner. PPDiff builds upon our developed **S**equence **S**tructure **I**nterleaving **N**etwork with **C**ausal attention layers (SSINC), which integrates interleaved self-attention layers to capture global amino acid correlations, $k$-nearest neighbor ($k$NN) equivariant graph layers to model local interactions in three-dimensional (3D) space, and causal attention layers to simplify the intricate interdependencies within the protein sequence. To assess PPDiff, we curate PPBench, a general protein-protein complex dataset comprising 706,360 complexes from the Protein Data Bank (PDB). The model is pretrained on PPBench and finetuned on two real-world applications: target-protein minibinder complex design and antigen-antibody complex design. PPDiff consistently surpasses baseline methods, achieving success rates of 50.00%, 23.16%, and 16.89% for the pretraining task and the two downstream applications, respectively.

## 1. Introduction

Designing proteins with high affinity and specificity to target proteins is important in biomedicine, with applications spanning therapeutic development (Nelson et al., 2010), diagnostics (Brennan et al., 2010) and imaging reagents (Stern et al., 2013). Empirical selection methods, such as screening high-complexity random libraries of antibodies (Chao et al., 2006) or alternative scaffolds (Hackel et al., 2008), have demonstrated the ability to generate binders for specific targets. While effective, these approaches require considerable experimental effort, making them both time-consuming and resource-intensive. Traditional computational methods leveraging physicochemical properties offer a more efficient alternative, enabling the design of binders by focusing on specific surface patches of the protein target (Chevalier et al., 2017; Silva et al., 2019). However, these approaches are often limited by the lack of obvious surface pockets on many target proteins, thereby restricting their applicability to a narrow set of binding interactions.

In parallel with advances in traditional computational binder design, deep learning methods have achieved remarkable accuracy in predicting and designing protein-protein interactions (Evans et al., 2021; Bennett et al., 2023; Zambaldi et al., 2024). These approaches have made it possible to design binders for specific targets without relying on high-throughput screening (Goudy et al., 2023). In certain cases, such as peptides (Vázquez Torres et al., 2024) and disordered targets (Wu et al., 2024), high binding affinity has been achieved without the need for extensive experimental optimization. Despite these successes, the overall success rate remains low due to two primary failure modes: the designed sequence may not fold into the intended structure, and the designed structure may not bind to the target protein. Moreover, most existing approaches are highly specialized for particular interaction types, leaving many target proteins intractable.

To address these challenges, we present PPDiff, a diffusion model building upon our developed **S**equence **S**tructure **I**nterleaving **N**etwork with **C**ausal attention layers (SSINC), to co-design the sequence and structure of binders for arbitrary protein targets in a non-autoregressive manner. In this framework, proteins are represented as residue point sets in 3D space, with each residue linked to a 3D Cartesian coordinate. The model employs a diffusion process for both continuous residue coordinates and discrete residue types,

---
[*]Work done during the internship at NEC Laboratories America. [1]Language Technologies Institute, Carnegie Mellon University [2]NEC Laboratories America. Correspondence to: Zhenqiao Song <zhenqiaosong@cmu.edu>.

*Proceedings of the 42$^{nd}$ International Conference on Machine Learning*, Vancouver, Canada. PMLR 267, 2025. Copyright 2025 by the author(s).

progressively adding noise to train a joint generative model. SSINC integrates interleaved self-attention layers, capturing global amino acid correlations, with $k$NN equivariant graph convolutional layers to model local 3D interactions. Causal attention layers are also introduced to simplify the intricate dependencies within the protein sequence. By simultaneously modeling sequence, structure, and their complex interdependencies, PPDiff mitigates primary failure modes in binder design while improving the diversity and novelty introduced by the diffusion process to enhance success rates.

Our contributions are listed as follows:

- We propose PPDiff, a diffusion model building upon the SSINC network to co-design the sequence and structure of binder proteins for arbitrary protein targets.
- We create PPBench, a general protein-protein complex dataset comprising 706,360 complexes curated from PDB.
- Our model pretrained on PPBench demonstrates a success rate of 50.00% on top-1 candidate, evaluated using a combination of metrics: ipTM, pTM, PAE, and pLDDT assessed by AlphaFold3 (Abramson et al., 2024). Notably, our model also achieves superior novelty and diversity scores, highlighting its ability to design novel and diverse protein-binding proteins with high success rates. The top-1 candidates are uploaded in the supplementary material.
- We finetune the pretrained PPDiff on two important real-world applications: target-protein mini-binder complex design and antigen-antibody complex design. The finetuned models achieve success rates of 23.16% and 16.89%, respectively, across these tasks, demonstrating its effectiveness in addressing significant challenges in biomedicine.

## 2. Related Work

**Methods for Protein-Protein Complex Design.** Designing proteins with high affinity and specificity for protein targets has been extensively explored using a wide range of approaches. The most commonly employed methods involve immunizing an animal with a target to induce antibody (Gray et al., 2020) or screening high-complexity random libraries of protein scaffolds (Chao et al., 2006; Hackel et al., 2008). While these approaches have proven effective, they are resource-intensive and require significant experimental effort. Traditional computational methods for binder design have sought to model physicochemical properties to accelerate the design process. Chevalier et al. (2017) introduce a massively parallel approach for designing, manufacturing, and screening mini-protein binders, with biophysical property characterization. Fleishman et al. (2011) develop a general computational framework for designing proteins that bind to specific surface patches of targets by

accounting for their physicochemical constraints. Cao et al. (2022) propose a general Rosetta-based approach to design binder proteins using only the structure of the target. While these approaches provide a systematic route for binder design, the reliance on a limited number of hotspot residues restricts them to a narrow range of interaction types. Recent advances in deep learning have significantly improved the prediction and design of protein-protein interactions (Evans et al., 2021; Humphreys et al., 2021; Bryant et al., 2022; Pacesa et al., 2024). These advancements have enabled the design of binders for certain targets without the need for high-throughput screening (Watson et al., 2023; Gainza et al., 2023; Goudy et al., 2023). Bennett et al. (2023) explore the augmentation of energy-based protein binder design with deep learning, while Zambaldi et al. (2024) introduce AlphaProteo for designing binders targeting eight different proteins. Despite these developments, the overall success rate for binder design remains low, with many protein targets still intractable (Yang et al., 2024; Berger et al., 2024).

**Diffusion Models.** Diffusion models (Sohl-Dickstein et al., 2015) have emerged as a prominent class of latent generative models. Ho et al. (2020) introduce denoising diffusion probabilistic models (DDPM), establishing a connection between diffusion models and denoising score-based models (Song & Ermon, 2019). Diffusion models have demonstrated remarkable success in generating high-quality images (Ho et al., 2020; Nichol & Dhariwal, 2021) and texts (Hoogeboom et al., 2021; Austin et al., 2021; Li et al., 2022). More recently, they have been applied to protein design. For instance, RFdiffusion (Watson et al., 2023) and FrameDiff (Yim et al., 2023) are continuous diffusion models capable of generating novel protein structures. EvoDiff (Alamdari et al., 2023) and DPLM (Wang et al., 2024) are discrete diffusion models designed for protein sequence generation. However, the application of diffusion models for the joint design of protein sequences and structures binding to specific protein targets still remains under-explored.

## 3. Proposed Method: PPDiff

### 3.1. Problem Definition

A protein consists of a chain of amino acids connected by peptide bonds, which folds into a proper 3D structure. Let $\mathcal{A}$ be the set of 20 common amino acids. Given a target protein $\mathcal{T}$ with its sequence of $M$ amino acids $s^{\mathcal{T}} = \{s_1^{\mathcal{T}}, s_2^{\mathcal{T}}, ..., s_M^{\mathcal{T}}\} \in \mathcal{A}^M$ and the 3D coordinates of the amino acids' alpha carbons $x^{\mathcal{T}} = [x_1^{\mathcal{T}}, x_2^{\mathcal{T}}, ..., x_M^{\mathcal{T}}]^T \in R^{M \times 3}$, the goal is to generate a protein $\mathcal{B}$ that binds to the target. That is to predict the amino acid sequence $s^{\mathcal{B}} = \{s_1^{\mathcal{B}}, s_2^{\mathcal{B}}, ..., s_N^{\mathcal{B}}\} \in \mathcal{A}^N$ and their alpha carbon ($C_\alpha$) coordinates $x^{\mathcal{B}} = [x_1^{\mathcal{B}}, x_2^{\mathcal{B}}, ..., x_N^{\mathcal{B}}]^T \in R^{N \times 3}$. $M$ and $N$ are the sequence lengths of the target and binder proteins.

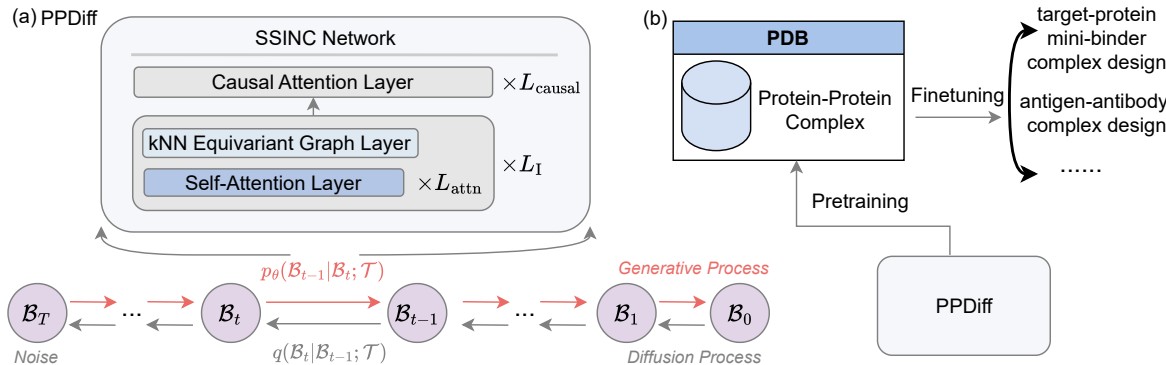

*Figure 1.* (a) Overall architecture of our proposed PPDiff. (b) We first pretrain PPDiff on PPBench, a general protein-protein complex dataset curated from PDB. Then we can finetune the pretrained model on important real-world protein-protein complex design applications, such as target-protein mini-binder complex design and antigen-antibody complex design.

We formulate the protein-protein complex design problem as learning the probabilistic generative model $p_\theta(\mathcal{B}|\mathcal{T})$ with parameters $\theta$. To this end, we propose a novel model PPDiff.

### 3.2. Overall Model Architecture

PPDiff aims to simultaneously generate the sequence and backbone structure of a binder protein for a given protein target. PPDiff is a denoising diffusion probabilistic model (DDPM) (Ho et al., 2020) with a joint sequence-structure generation network (Figure 1(a)). The model comprises a forward *diffusion* process to add noises and a reverse *generative* process, both defined as Markov chains. The diffusion process gradually perturbs the ground-truth data $\mathcal{B}_0 \sim q(\mathcal{B}_0|\mathcal{T})$ into a stationary distribution $\mathcal{B}_T \sim q_{\text{noise}}$ with $T$ increasingly noisy steps:

$$q(\mathcal{B}_{1:T}|\mathcal{B}_0, \mathcal{T}) = \Pi_{t=1}^T q(\mathcal{B}_t|\mathcal{B}_{t-1}, \mathcal{T}) \quad (1)$$

where $\mathcal{B}_1, ..., \mathcal{B}_T$ is a sequence of latent variables, each contains both amino acid sequence $s_t^\mathcal{B}$ and alpha carbon coordinates $x_t^\mathcal{B}$. Notice that it is a big challenge to model the latent variables because $\mathcal{B}_t$ contains both discrete and continuous components. The generative process parameterizes the transition kernel $p_\theta(\mathcal{B}_{t-1}|\mathcal{B}_t, \mathcal{T})$ to recover the data distribution from the noise distribution:

$$p_\theta(\mathcal{B}_{0:T}|\mathcal{T}) = p(\mathcal{B}_T)\Pi_{t=1}^T p_\theta(\mathcal{B}_{t-1}|\mathcal{B}_t, \mathcal{T}) \quad (2)$$

To fit the model $p_\theta(\mathcal{B}_0|\mathcal{T})$ to the data distribution $q(\mathcal{B}_0|\mathcal{T})$, the denoising model is typically optimized by the variational bound of the log-likelihood (Ho et al., 2020):

$$E_{q(\mathcal{B}_0|\mathcal{T})}[\log p_\theta(\mathcal{B}_0|\mathcal{T})] \geq E_{q(\mathcal{B}_{0:T}|\mathcal{T})}[\log \frac{p_\theta(\mathcal{B}_{0:T}|\mathcal{T})}{q(\mathcal{B}_{1:T}|\mathcal{B}_0, \mathcal{T})}]$$

$$= E_{q(\mathcal{B}_0|\mathcal{T})}\big[\log p_\theta(\mathcal{B}_0|\mathcal{B}_1, \mathcal{T}) + \text{const}$$

$$+ \sum_{t=2}^T \underbrace{-\text{KL}[q(\mathcal{B}_{t-1}|\mathcal{B}_t, \mathcal{B}_0, \mathcal{T})||p_\theta(\mathcal{B}_{t-1}|\mathcal{B}_t, \mathcal{T})]}_{\mathcal{L}_t}\big]$$

$$\quad (3)$$

To reduce the failure modes of previous methods and enhance the consistency between the designed binder sequence and structure, PPDiff is designed to jointly generate both the discrete residue sequence and the continuous backbone structure. This requires implementing diffusion and generative processes that operate on both discrete residue types and continuous spatial coordinates. In the following subsections, we will provide a detailed introduction of the diffusion and generative processes in PPDiff, followed by a discussion of the learning objective and the informative guidance strategy used to generate high-quality candidates during inference.

### 3.3. PPDiff Diffusion Process

Following Guan et al. (2023), we model the joint diffusion distribution of protein residue sequences $s_t^\mathcal{B}$ and backbone structures $x_t^\mathcal{B}$ independently. Such a formulation allows for efficient sampling of noisy data for both modalities, i.e. discrete residue types and continuous coordinates. For the discrete protein sequence $s^\mathcal{B}$, we adopt a categorical distribution $\text{Cat}(s^\mathcal{B}; p)$ as proposed by Hoogeboom et al. (2021), where $p$ represents a vector on the $(|\mathcal{A}| - 1 = 19)$-dimensional probability simplex. For the continuous backbone structure, we follow Ho et al. (2020) and use a Gaussian distribution $\mathcal{N}$ to model spatial coordinates. At each diffusion step $t$, uniform noise is added to the residue types across all categories, and Gaussian noise is applied to the structural coordinates. This process follows a Markov chain with a predefined schedule $\beta_1, \beta_2, ..., \beta_T$ as follows:

$$q(\mathcal{B}_t|\mathcal{B}_{t-1}, \mathcal{T}) = q(s_t^\mathcal{B}|s_{t-1}^\mathcal{B}, \mathcal{T}) \cdot q(x_t^\mathcal{B}|x_{t-1}^\mathcal{B}, \mathcal{T})$$

$$q(s_t^\mathcal{B}|s_{t-1}^\mathcal{B}, \mathcal{T}) = \text{Cat}(s_t^\mathcal{B}; (1 - \beta_t)s_{t-1}^\mathcal{B} + \beta_t/K) \quad (4)$$

$$q(x_t^\mathcal{B}|x_{t-1}^\mathcal{B}, \mathcal{T}) = \mathcal{N}(x_t^\mathcal{B}; \sqrt{1 - \beta_t}x_{t-1}^\mathcal{B}, \beta_t I)$$

In practice, the schedules for different data modalities can be different.

Denoting $\alpha_t = 1 - \beta_t$ and $\overline{\alpha}_t = \Pi_{i=1}^t \alpha_i$, we can calculate the noisy data distributions $q(s_t^\mathcal{B}|s_0^\mathcal{B}, \mathcal{T})$ and $q(x_t^\mathcal{B}|x_0^\mathcal{B}, \mathcal{T})$

in closed-form:

$$q(\boldsymbol{s}_t^{\mathcal{B}}|\boldsymbol{s}_0^{\mathcal{B}}, \mathcal{T}) = \text{Cat}(\boldsymbol{s}_t^{\mathcal{B}}; \overline{\alpha}_t \boldsymbol{s}_0^{\mathcal{B}} + (1 - \overline{\alpha}_t)/K)$$
$$q(\boldsymbol{x}_t^{\mathcal{B}}|\boldsymbol{x}_0^{\mathcal{B}}, \mathcal{T}) = \mathcal{N}(\boldsymbol{x}_t^{\mathcal{B}}; \sqrt{\overline{\alpha}_t}\boldsymbol{x}_0^{\mathcal{B}}, (1 - \overline{\alpha}_t)\boldsymbol{I})$$

(5)

Based on Eq. 4 and Eq. 5, we can calculate the posterior distributions $q(\boldsymbol{s}_{t-1}^{\mathcal{B}}|\boldsymbol{s}_t^{\mathcal{B}}, \boldsymbol{s}_0^{\mathcal{B}}, \mathcal{T})$ and $q(\boldsymbol{x}_{t-1}^{\mathcal{B}}|\boldsymbol{x}_t^{\mathcal{B}}, \boldsymbol{x}_0^{\mathcal{B}}, \mathcal{T})$ using Bayes rule in closed-form:

$$q(\boldsymbol{s}_{t-1}^{\mathcal{B}}|\boldsymbol{s}_t^{\mathcal{B}}, \boldsymbol{s}_0^{\mathcal{B}}, \mathcal{T}) = \text{Cat}(\boldsymbol{s}_{t-1}^{\mathcal{B}}; \boldsymbol{\theta}_{\text{post}}^{\boldsymbol{s}}(\boldsymbol{s}_t^{\mathcal{B}}, \boldsymbol{s}_0^{\mathcal{B}}))$$

$$\boldsymbol{\theta}_{\text{post}}^{\boldsymbol{s}}(\boldsymbol{s}_t^{\mathcal{B}}, \boldsymbol{s}_0^{\mathcal{B}}) = \tilde{\boldsymbol{\theta}}^{\boldsymbol{s}} / \sum_{k=1}^{|\mathcal{A}|} \tilde{\theta}_k^{\boldsymbol{s}}$$

$$\tilde{\boldsymbol{\theta}}^{\boldsymbol{s}} = [\alpha_t \boldsymbol{s}_t^{\mathcal{B}} + (1 - \alpha_t)/K] \odot [\overline{\alpha}_{t-1}\boldsymbol{s}_0^{\mathcal{B}} + (1 - \overline{\alpha}_{t-1})/K]$$

$$q(\boldsymbol{x}_{t-1}^{\mathcal{B}}|\boldsymbol{x}_t^{\mathcal{B}}, \boldsymbol{x}_0^{\mathcal{B}}, \mathcal{T}) = \mathcal{N}(\boldsymbol{x}_{t-1}^{\mathcal{B}}; \tilde{\boldsymbol{\mu}}_t(\boldsymbol{x}_t^{\mathcal{B}}, \boldsymbol{x}_0^{\mathcal{B}}), \tilde{\beta}_t \boldsymbol{I})$$

$$\tilde{\boldsymbol{\mu}}_t(\boldsymbol{x}_t^{\mathcal{B}}, \boldsymbol{x}_0^{\mathcal{B}}) = \frac{\sqrt{\overline{\alpha}_{t-1}}\beta_t}{1 - \overline{\alpha}_t}\boldsymbol{x}_0^{\mathcal{B}} + \frac{\sqrt{\alpha_t}(1 - \overline{\alpha}_{t-1})}{1 - \overline{\alpha}_t}\boldsymbol{x}_t^{\mathcal{B}}$$

$$\tilde{\beta}_t = \frac{1 - \overline{\alpha}_{t-1}}{1 - \overline{\alpha}_t}\beta_t$$

(6)

### 3.4. The SSINC Network for PPDiff Generative Process

The generative process recovers the data distribution from the noise distribution. We parameterize the reverse generative process using our developed SSINC network by $\theta$:

$$p_\theta(\mathcal{B}_{t-1}|\mathcal{B}_t, \mathcal{T}) = p_\theta(\boldsymbol{s}_{t-1}^{\mathcal{B}}|\boldsymbol{s}_t^{\mathcal{B}}, \boldsymbol{x}_t^{\mathcal{B}}, \mathcal{T}) \cdot p_\theta(\boldsymbol{x}_{t-1}^{\mathcal{B}}|\boldsymbol{x}_t^{\mathcal{B}}, \boldsymbol{s}_t^{\mathcal{B}}, \mathcal{T})$$
$$p_\theta(\boldsymbol{s}_{t-1}^{\mathcal{B}}|\boldsymbol{s}_t^{\mathcal{B}}, \boldsymbol{x}_t^{\mathcal{B}}, \mathcal{T}) = \text{Cat}(\boldsymbol{s}_{t-1}^{\mathcal{B}}; \boldsymbol{\theta}_{\text{post}}^{\boldsymbol{s}}(\boldsymbol{s}_t^{\mathcal{B}}, \hat{\boldsymbol{s}}_0^{\mathcal{B}}))$$
$$p_\theta(\boldsymbol{x}_{t-1}^{\mathcal{B}}|\boldsymbol{x}_t^{\mathcal{B}}, \boldsymbol{s}_t^{\mathcal{B}}, \mathcal{T}) = \mathcal{N}(\boldsymbol{x}_{t-1}^{\mathcal{B}}; \boldsymbol{\mu}_\theta(\boldsymbol{s}_t^{\mathcal{B}}, \boldsymbol{x}_t^{\mathcal{B}}, t, \mathcal{T}), \sigma_t^2 \boldsymbol{I})$$

(7)

The SSINC network is composed of $L_I$ interleaved blocks, each containing $L_{\text{attn}}$ self-attention layers (Vaswani, 2017) to capture global correlations among amino acids and a $k$NN equivariant graph convolutional layer (Satorras et al., 2021) to model local interactions among neighboring residues in 3D space, as illustrated in Figure 1 (a). In the $l$-th interleaved block, the representations are calculated as follows:

$$\boldsymbol{H}_t^{l+0.5} = \text{Self-Attention}(\boldsymbol{H}_t^l; \mathcal{T})$$
$$\boldsymbol{m}_{t,ik}^{l+0.5} = f_m([\boldsymbol{h}_{t,i}^{l+0.5}; \boldsymbol{h}_{t,k}^{l+0.5}; ||\boldsymbol{x}_{t,i}^l - \boldsymbol{x}_{t,k}^l||_2]; \mathcal{T})$$
$$w_{t,ik}^{l+0.5} = \text{Softmax}(\boldsymbol{m}_{t,ik}^{l+0.5}), \boldsymbol{m}_{t,ik}^{l+1} = w_{t,ik}^{l+0.5} \cdot \boldsymbol{m}_{t,ik}^{l+0.5}$$
$$\boldsymbol{c}_i^{l+1} = \sum_{k \in \text{N(i)}} \boldsymbol{m}_{t,ik}^{l+1}, \text{weight} = \sigma(f_w(\boldsymbol{c}_i^{l+1}))$$
$$\boldsymbol{h}_{t,i}^{l+1} = \boldsymbol{h}_{t,i}^{l+0.5} + \sum_{k \in \text{N(i)}} \text{weight} \cdot \boldsymbol{m}_{t,ik}^{l+1}$$
$$\boldsymbol{x}_{t,i}^{l+1} = \boldsymbol{x}_{t,i}^l + \sum_{k \in \text{N(i)}} (\boldsymbol{x}_{t,i}^l - \boldsymbol{x}_{t,k}^l) \cdot f_x(\boldsymbol{m}_{t,ik}^{l+1})$$

(8)

where $\boldsymbol{H}_t^l = [\boldsymbol{h}_{t,1}^l, \boldsymbol{h}_{t,2}^l, ..., \boldsymbol{h}_{t,N}^l]^T$ denotes the residue representation matrix at the $l$-th block and time step $t$, while $\boldsymbol{H}_t^0 = [\boldsymbol{s}_1^{\mathcal{B}}, \boldsymbol{s}_2^{\mathcal{B}}, ..., \boldsymbol{s}_N^{\mathcal{B}}]^T$ is the embedding matrix of binder protein sequence. Functions $f_*$ denote feed-forward layers and $\sigma$ denotes sigmoid function. $N(i)$ is the set of $k$-nearest neighbors of the $i$-th residue. Finally, $\boldsymbol{x}_t^{L_I}$ is the predicted

structure at time step $t$, i.e. $\hat{\boldsymbol{x}}_0^{\mathcal{B}}$. To make the learning easier, $L_{\text{causal}}$ causal attention layers are added on top of the SSINC network to capture the sequential dependencies:

$$\boldsymbol{h}_{t,i}^{\text{out}} = \text{Causal-Attention}\left(\boldsymbol{h}_{t,i}^{L_I}, \boldsymbol{H}_{t,1:i}^{L_I}\right)$$
$$\hat{\boldsymbol{s}}_0^{\mathcal{B}} = \text{Softmax}(\boldsymbol{H}_t^{\text{out}})$$

(9)

where the $i$-th residue can only attend to its previous residues $\boldsymbol{H}_{t,1:i}^{L_I}$. We can then approximate the posterior distributions $p_\theta(\boldsymbol{s}_{t-1}^{\mathcal{B}}|\boldsymbol{s}_t^{\mathcal{B}}, \boldsymbol{x}_t^{\mathcal{B}}, \mathcal{T})$ and $p_\theta(\boldsymbol{x}_{t-1}^{\mathcal{B}}|\boldsymbol{x}_t^{\mathcal{B}}, \boldsymbol{s}_t^{\mathcal{B}}, \mathcal{T})$ using the predicted $\hat{\boldsymbol{s}}_0^{\mathcal{B}}$ and $\hat{\boldsymbol{x}}_0^{\mathcal{B}}$.

### 3.5. Training Objective

We learn the model by maximizing the variational lower bound of the log-likelihood defined in Eq. 3, where $\mathcal{L}_t$ is calculated as:

$$\mathcal{L}_t = -\text{KL}[q(\boldsymbol{s}_{t-1}^{\mathcal{B}}|\boldsymbol{s}_t^{\mathcal{B}}, \boldsymbol{s}_0^{\mathcal{B}}, \mathcal{T})||p_\theta(\boldsymbol{s}_{t-1}^{\mathcal{B}}|\boldsymbol{s}_t^{\mathcal{B}}, \boldsymbol{x}_t^{\mathcal{B}}, \mathcal{T})]$$
$$- \text{KL}[q(\boldsymbol{x}_{t-1}^{\mathcal{B}}|\boldsymbol{x}_t^{\mathcal{B}}, \boldsymbol{x}_0^{\mathcal{B}}, \mathcal{T})||p_\theta(\boldsymbol{x}_{t-1}^{\mathcal{B}}|\boldsymbol{x}_t^{\mathcal{B}}, \boldsymbol{s}_t^{\mathcal{B}}, \mathcal{T})]$$
$$= -\underbrace{\left\{\sum_k \boldsymbol{\theta}_{\text{post}}^{\boldsymbol{s}}(\boldsymbol{s}_t^{\mathcal{B}}, \boldsymbol{s}_0^{\mathcal{B}})_k \log \frac{\boldsymbol{\theta}_{\text{post}}^{\boldsymbol{s}}(\boldsymbol{s}_t^{\mathcal{B}}, \boldsymbol{s}_0^{\mathcal{B}})_k}{\boldsymbol{\theta}_{\text{post}}^{\boldsymbol{s}}(\boldsymbol{s}_t^{\mathcal{B}}, \hat{\boldsymbol{s}}_0^{\mathcal{B}})_k}\right\}}_{\mathcal{L}_t^{\boldsymbol{s}}}$$
$$- \underbrace{\left\{\frac{1}{2\sigma_t^2}||\tilde{\boldsymbol{\mu}}_t(\boldsymbol{x}_t^{\mathcal{B}}, \boldsymbol{x}_0^{\mathcal{B}}) - \boldsymbol{\mu}_\theta(\boldsymbol{s}_t^{\mathcal{B}}, \boldsymbol{x}_t^{\mathcal{B}}, t, \mathcal{T})||^2 + \text{const}\right\}}_{\mathcal{L}_t^{\boldsymbol{x}}}$$

(10)

where $\boldsymbol{\theta}_{\text{post}}^{\boldsymbol{s}}(\boldsymbol{s}_t^{\mathcal{B}}, \hat{\boldsymbol{s}}_0^{\mathcal{B}})$ is defined in Eq. 6, with $\hat{\boldsymbol{s}}_0^{\mathcal{B}}$ calculated in Eq. 9. $\mathcal{L}_t^{\boldsymbol{x}}$ can be further symplified as:

$$\mathcal{L}_t^{\boldsymbol{x}} = -\left\{\lambda_t||\boldsymbol{x}_0^{\mathcal{B}} - \hat{\boldsymbol{x}}_0^{\mathcal{B}}||^2 + \text{const}\right\}, \quad \lambda_t = \frac{\overline{\alpha}_{t-1}\beta_t^2}{2\sigma^2(1 - \overline{\alpha}_t)^2}$$

(11)

Following previous work (Guan et al., 2023), we set $\lambda_t = 1$. Furthermore, $\log p_\theta(\mathcal{B}_0|\mathcal{B}_1, \mathcal{T})$ can be calculated as :

$$\log p_\theta(\mathcal{B}_0|\mathcal{B}_1, \mathcal{T}) = \log p_\theta(\boldsymbol{s}_0^{\mathcal{B}}|\boldsymbol{s}_1^{\mathcal{B}}, \boldsymbol{x}_1^{\mathcal{B}}, \mathcal{T})$$
$$+ \log p_\theta(\boldsymbol{x}_0^{\mathcal{B}}|\boldsymbol{x}_1^{\mathcal{B}}, \boldsymbol{s}_1^{\mathcal{B}}, \mathcal{T})$$
$$= \sum_k \boldsymbol{s}_{0,k}^{\mathcal{B}} \log \hat{\boldsymbol{s}}_{0,k}^{\mathcal{B}} - ||\boldsymbol{x}_0^{\mathcal{B}} - \hat{\boldsymbol{x}}_0^{\mathcal{B}}||^2$$

(12)

### 3.6. Informative Guidance for Inference

As suggested by previous studies (Graikos et al., 2022; Wu et al., 2022), an informative prior distribution can significantly enhance model performance. To leverage this insight, we introduce an informative guidance to select better starting points rather than random noise. For backbone structure guidance, we define the following energy function:

$$E_{\text{knn}}(\mathcal{B}) = \sum_{i=1}^N (\text{knn-dist}(\boldsymbol{x}_i) - \mu_{\text{knn}})^2$$

(13)

where knn-dist$(\boldsymbol{x}_i) = \frac{1}{k} \sum_{j \in N(i)} ||\boldsymbol{x}_i - \boldsymbol{x}_j||^2$ denotes the average distance from $i$-th residue to its $k$ nearest neighbors and $\mu_{\text{knn}}$ is the empirical mean of knn-dist$(\boldsymbol{x})$ across all residues in the training dataset. In our setting, $k$ is set to 4. During inference, we sample 10 noisy structures and choose the one with the lowest energy. For sequence guidance, we randomly sample secondary structure fragments from the training dataset, identified by using DSSP (Kabsch & Sander, 1983). This strategy ensures the initialized structures and sequences maintain geometric similarities to the training data, providing the model with a more informed starting point and enhancing the quality of candidate generation.

## 4. Experiments

In this section, we first describe our PPBench construction process in Section 4.1 and experimental setup in Section 4.2. Then we conduct extensive experiments on a **General Protein-Protein Complex Design** task (Section 4.3) and two real-world applications, **Target-Protein Mini-Binder Complex Design** (Section 4.4) and **Antigen-Antibody Complex Design** (Section 4.5). The specific experimental settings are introduced in each task section.

### 4.1. PPBench Construction

Our goal is to develop a general protein-protein complex design model that serves as a foundational model, adaptable for finetuning in diverse real-world protein-protein interaction applications. To achieve this, we curate a large-scale protein-protein complex dataset by identifying chain-pair interfaces, following the data processing pipeline in AlphaFold3 (AF3). Details of the data curation process are provided in Appendix C.1. This process results in 367,016 chain-pair interfaces. Treating each chain in a chain-pair interface as a potential target protein, we gather a total of 734,032 complexes. Protein sequences are then clustered based on 50% sequence identity, producing 22,847 clusters. To prepare the dataset for training and evaluation, we designate 10 clusters each for validation and testing, with the remaining clusters reserved for training. To ensure efficient processing, training data are further filtered to exclude sequences longer than 1,024 residues, while validation and testing data are restricted to sequences no longer than 512 residues, finally resulting in a total of 706,360 complexes. Detailed data statistics are provided in the Appendix A.1.

### 4.2. Experimental Setup

**Implementation Details.** Our architecture incorporates three interleaving blocks and one causal attention layer. Each interleaving block comprises 11 self-attention layers and one $k$NN-based equivariant graph convolutional layer. The parameters for self-attention layers are initialized using the pretrained 650M ESM-2 model (Lin et al., 2022). PPDiff has a total of 692M parameters. The diffusion process is configured with 1,000 steps, employing a cosine schedule for sequence diffusion and a sigmoid schedule for structure diffusion. Additional model training details are provided in Appendix C.2.

**Baseline Models.** Given the absence of large-scale models for protein-protein complex sequence and structure co-design, we evaluate the effectiveness of PPDiff by comparing it with several representative baselines: (1) **SEnc+ProteinMPNN** first employs a continuous diffusion model based on structured encoder (Ingraham et al., 2019) enhanced with spatial features to design complex backbone structures, followed by ProteinMPNN (Dauparas et al., 2022) to predict binder protein sequences based on the designed backbone structure. (2) **InterleavingDiff** is a variant of PPDiff without the casual attention layers, of which the backbone model is the interleaving network (Song et al., 2024b). (3) **SSINC Network** has the same architecture as PPDiff, but it is trained without diffusion process. To ensure a fair comparison, all models are trained on the curated PPBench using their official implementations.

### 4.3. General Protein-Protein Complex Design

This task aims to develop a general protein-protein complex design foundational model that can be readily finetuned for diverse real-world protein-protein interaction applications (Figure 1 (b)). To this end, we pretrain our proposed PPDiff on the curated PPBench.

**Evaluation Metrics.** We evaluate the designed protein-protein complexes based on structure stability, reliability, and functional interaction using AF3 metrics. Specifically, **pLDDT** assesses structural stability, **pTM** and **PAE** measure structural reliability, and **ipTM** evaluates interface interactions, reflecting functionality. Detailed explanations of these metrics are provided in Appendix C.3. As per prior studies (Binder et al., 2022; Bennett et al., 2023; Abramson et al., 2024), a successful complex satisfies ipTM$\geq$0.8, pTM$\geq$0.8, PAE$\leq$10 and pLDDT$\geq$80. For each target protein $\mathcal{T}$ in the test set $\mathcal{D}_{\text{test}}$, we generate 100 binder candidates and rank them by their complex pLDDT scores, selecting the top-$k$ candidates $\mathcal{C}$. The **success rate (SR)** for the top-$k$ candidates is:

$$\text{SR} = \frac{1}{|\mathcal{D}_{\text{test}}| * k} \sum_{\mathcal{T} \in \mathcal{D}_{\text{test}}} \sum_{i=1}^{k} \text{II}(f(\mathcal{T}, \mathcal{B}_i)) \quad (14)$$

where $f(\mathcal{T}, \mathcal{B}_i)$ ensures if the thresholds for ipTM, pTM, PAE, and pLDDT are met, and $\text{II}(\text{true}) = 1$, $\text{II}(\text{false}) = 0$. Here, $\mathcal{B}_i$ is the $i$-th candidate. We also evaluate **diversity** and **novelty** scores. Diversity quantifies sequence variability among the top-$k$ candidates, while novelty measures the

| | Methods | ipTM (↑) | pTM (↑) | PAE (↓) | pLDDT(↑) | Success Rate (↑) | Novelty (↑) | Diversity (↑) |
|---|---|---|---|---|---|---|---|---|
| | Ground Truth | 0.691 | 0.782 | 7.901 | 86.987 | 69.64% | – | – |
| top-1 | SEnc +ProteinMPNN | 0.629 | 0.713 | 10.374 | 79.819 | 35.08% | 59.76% | – |
| | InterleavingDiff | 0.674 | 0.757 | 9.934 | 82.411 | 45.61% | 89.45% | – |
| | SSINC Network | 0.660 | 0.751 | 9.713 | 82.998 | 47.36% | 37.46% | – |
| | PPDiff | **0.700** | **0.779** | **9.153** | **83.765** | **50.00%** | **89.46%** | – |
| top-5 | SEnc +ProteinMPNN | 0.617 | 0.682 | 11.789 | 78.021 | 29.82% | 61.78% | 58.72% |
| | InterleavingDiff | 0.648 | 0.732 | 11.010 | 79.944 | 37.85% | **90.66%** | **91.82%** |
| | SSINC Network | 0.647 | 0.745 | **9.903** | **82.290** | 43.85% | 37.49% | 15.12% |
| | PPDiff | **0.665** | **0.747** | 10.231 | 81.659 | **45.71%** | 88.93% | 90.76% |
| top-10 | SEnc +ProteinMPNN | 0.582 | 0.671 | 12.347 | 76.892 | 21.05% | 62.98% | 58.93% |
| | InterleavingDiff | 0.629 | 0.716 | 11.745 | 78.328 | 24.28% | **90.64%** | **91.89%** |
| | SSINC Network | 0.620 | 0.721 | **10.075** | **81.852** | 36.84% | 37.45% | 15.03% |
| | PPDiff | **0.633** | **0.729** | 10.895 | 80.322 | **37.68%** | 89.10% | 91.09% |

*Table 1.* Model performance on general protein-protein complex design task. (↑): the higher the better. (↓): the lower the better. Calculating the diversity score for the top-1 candidate is unnecessary, as it consists of only a single candidate. PPDiff consistently achieves the highest success rates for top-1, top-5, and top-10 candidates out of 100 candidates for each target protein.

deviation of candidates from the ground-truth sequence:

$$\text{diversity} = \frac{1}{(k*(k-1))} \sum_{i,j\in\mathcal{C}, i\neq j} 1 - \text{AAR}(s_i^{\mathcal{B}}, s_j^{\mathcal{B}})$$
$$\text{novelty} = \frac{1}{k} \sum_{i\in\mathcal{C}} 1 - \text{AAR}(s_i^{\mathcal{B}}, s^{\mathcal{B}})$$
(15)

where $s_i^{\mathcal{B}}$ is the sequence of $\mathcal{B}_i$, $s^{\mathcal{B}}$ is the ground truth binder sequence, and AAR denotes the amino acid recovery rate. The final diversity and novelty scores are averaged across the whole test set.

**Main Results.** The results for top-1, top-5, and top-10 candidates are reported in Table 1. **These results demonstrate that PPDiff excels in designing high-quality, diverse, and novel protein-binding proteins across a wide range of protein targets.** For the top-1 candidate, PPDiff achieves the best performance across all metrics. While PPDiff performs slightly worse than the SSINC Network on PAE and pLDDT for top-5 and top-10 candidates, it still achieves the highest success rate and significantly performs better in diversity and novelty. Although the diversity and novelty scores for PPDiff's top-5 and top-10 candidates are marginally lower than those of InterleavingDiff, PPDiff produces designs of much higher quality, validating the effectiveness of the causal attention layer.

### 4.4. Target-Protein Mini-Binder Complex Design

In this section, we aim to use PPDiff to design proteins that bind to specific protein targets with high affinity.

**Datasets.** We collect experimentally confirmed positive target-protein mini-binder complexes against ten targets with diverse structural properties from Bennett et al. (2023). For categories containing more than 50 complexes, an 8:1:1 random split is applied for training, validation, and test sets. For smaller categories, all complexes are included in the test

set to create a zero-shot evaluation scenario. Detailed data statistics are provided in Appendix A.2.

**Evaluation Metrics.** Following prior works (Bennett et al., 2023; Watson et al., 2023; Song et al., 2024a), we evaluate the binding affinities between the designed binders and target proteins using the **AlphaFold2 (AF2) pAE_interaction**[1] developed by Bennett et al. (2023). Their work demonstrates that AF2 pAE_interaction effectively differentiates experimentally validated binders from non-binders, achieving success rates ranging from 1.5% and 7% for target proteins FGFR2, IL7Ra, TrkA, InsulinR, PDGFR, and VirB8. Based on the calculated pAE_interaction score, we determine the **success rate** by generating five mini-binder candidates for each target protein. The success rate is defined as the proportion of designed binders that achieve a lower pAE_interaction score with the target protein than the ground truth positive binder. More evaluation details are provided in Appendix C.4.

**Main Results.** We finetune PPDiff on the target-protein mini-binder complex design task and the success rates are summarized in Table 2. **Our PPDiff achieves a superior success rate compared to all baselines, demonstrating its effectiveness in designing high-affinity binders.** However, we observe that all models fail to design successful binders for EGFR, likely due to the lack of deep pockets or distinct features in its binding sites (Callaway, 2024).

### 4.5. Antigen-Antibody Complex Design

In this section, we apply PPDiff to design high-quality antibodies binding to given antigens.

**Datasets.** Following Jin et al. (2022), we curate antigen-antibody complexes from the Structural Antibody

---
[1]https://github.com/nrbennet/dl_binder_design

| Models | Seen Class | | | | | Zero-Shot | | | | | Average |
|---|---|---|---|---|---|---|---|---|---|---|---|
| | FGFR2 | InsulinR | PDGFR | TGFb | VirB8 | H3 | IL7Ra | EGFR | TrkA | Tie2 | |
| SEnc+ProteinMPNN | 8.07% | 6.08% | 4.61% | 15.56% | 11.42% | 4.73% | 17.14% | 0.0 | 5.00% | 0.0 | 7.12% |
| InterleavingDiff | 10.00% | 6.95% | 11.53% | 35.56% | 8.57% | 48.42% | 34.28% | 0.0 | 10.00% | 10.00% | 19.32% |
| SSINC Network | **21.05%** | 0.0 | 3.07% | 20.00% | 2.85% | 7.36% | 22.85% | 0.0 | 5.00% | 0.0 | 10.96% |
| PPDiff | 7.36% | **10.43%** | **14.61%** | **35.56%** | **11.42%** | **55.26%** | **60.00%** | 0.0 | **20.00%** | **30.00%** | **23.16%** |

Table 2. The success rate (%, ↑) for all models on the target-protein mini-binder complex design task. "Average" refers to the overall success rate across the entire test set, rather than a simple mean across target categories. Our proposed PPDiff demonstrates a significant improvement in success rate, outperforming all previous methods by a substantial margin.

| Methods | ipTM (↑) | pTM (↑) | PAE (↓) | pLDDT(↑) | Success Rate (↑) | H1 Novelty (↑) | H2 Novelty (↑) | H3 Novelty (↑) |
|---|---|---|---|---|---|---|---|---|
| SEnc +ProteinMPNN | 0.448 | 0.578 | 14.418 | 77.786 | 6.24% | **61.78%** | 59.53% | 70.85% |
| InterleavingDiff | 0.496 | 0.653 | 13.583 | 80.939 | 12.00% | 57.54% | 63.59% | 75.58% |
| SSINC Network | 0.539 | 0.654 | **12.064** | 82.706 | 16.74% | 30.44% | 37.70% | 59.86% |
| PPDiff | **0.541** | **0.668** | 12.999 | **82.827** | **16.89%** | 57.79% | **66.39%** | **76.17%** |

Table 3. Model performance on antigen-antibody complex design task. (↑): the higher the better. (↓): the lower the better. Novelty scores for CDR-H1, CDR-H2, and CDR-H3 are denoted as H1, H2, and H3 novelty, respectively. PPDiff demonstrates superior performance, achieving the highest average success rate across the designed CDRs.

Database (Dunbar et al. (2014), SAbDab), resulting in 4,261 valid complexes after removing cases lacking a light chain or antigen. The dataset is split into training, validation, and test sets based on the clustering of CDRs. Taking CDR-H3 as an example, we use MMseqs2 (Steinegger & Söding, 2017) to cluster CDR-H3 sequences at 40% sequence identity, calculated using the BLOSUM62 substitution matrix (Henikoff & Henikoff, 1992). Clusters are then randomly divided into training, validation, and test sets in an 8:1:1 ratio. The same procedure is applied to create splits for CDR-H1 and CDR-H2, resulting in 1,034, 1,418, and 2,254 clusters, respectively. Details of the training, validation and test set sizes for each CDR clustering are provided in the Appendix A.3.

**Evaluation Metrics.** To guarantee a comprehensive evaluation of the designed antigen-antibody complex, we use the **ipTM, pTM, PAE, pLDDT, success rate and novelty** metrics introduced in the general protein-protein complex design evaluation (Section 4.3).

**Main Results.** We finetune the pretrained PPDiff on each of the antigen-antibody complex dataset derived from the clustering process and present the average performance in Table 3. Detailed results for each CDR clustering are given in Appendix B.1. **PPDiff achieves the highest antigen-antibody binding affinity (ipTM), success rate, and novelty scores.** It demonstrates PPDiff's ability to design high-affinity and novel antibodies for given antigens, highlighting its potential significance for real-world applications.

## 5. Analysis: Diving Deep into PPDiff

### 5.1. How Does The Causal Attention Layer Function?

To assess the influence of causal attention layers, we train the model with varying layer size (0 to 4) on PPBench. For each setting, we generate one candidate and evaluate the resulting complexes using AF3 with the same seed. The results, visualized in Figure 2 (a), illustrate that introducing causal attention layers significantly enhances design quality. However, increasing the number of layers above one does not provide additional benefits. Thus, **using a single causal attention layer achieves the optimal design quality**.

### 5.2. Effect of Diffusion Steps on Model Performance

To explore the impact of diffusion steps on PPDiff's performance, we train the model with different diffusion steps ranging from 500 to 2,000. For each trained model, a single candidate is generated and evaluated using AF3 under the same seed. The results, presented in Figure 2 (b), demonstrate that **PPDiff performs best when configured with 1,000 diffusion steps**.

### 5.3. Does Model Scale Help?

**PPDiff is scalable.** To evaluate the scalability of PPDiff, we compare the performance of our PPDiff (692M) with a version comprising 30 self-attention layers (164M). For each trained model, a single candidate is generated and evaluated using AF3 under the same seed. Figure 2 (e) shows the ipTM and pTM scores. The results reveal that the scaled-up PPDiff achieves significantly higher ipTM and pTM scores, highlighting the scalability and enhanced performance of our approach as the model size increases.

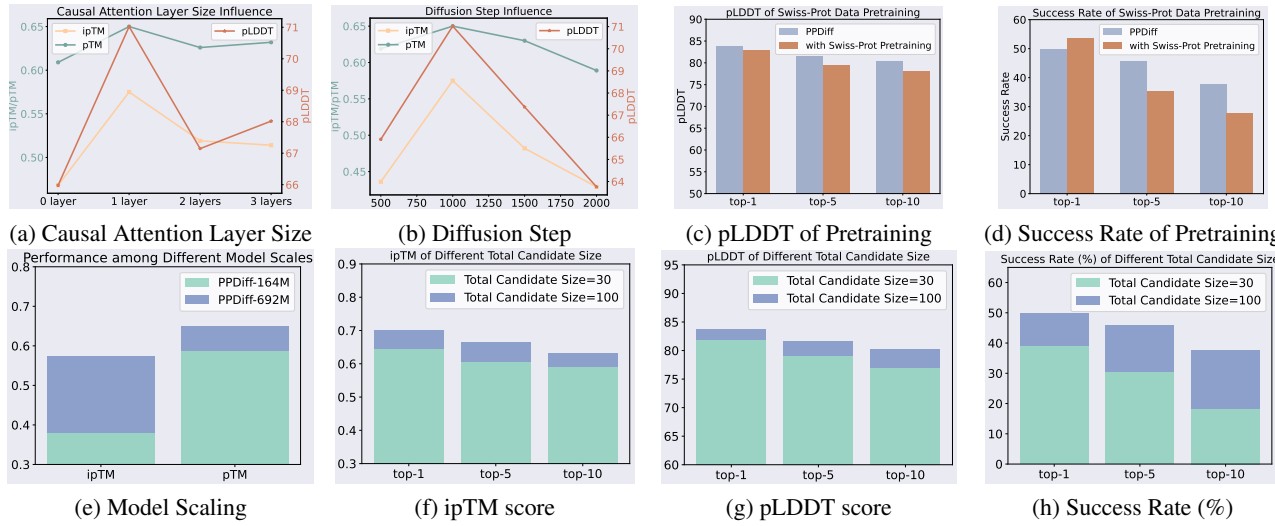

*Figure 2.* Ablation study on: (a) causal attention layer size, (b) different diffusion steps, (c-d) additional Swiss-Prot data pretraining, (e) different model scales, (f-h) different total candidate sizes.

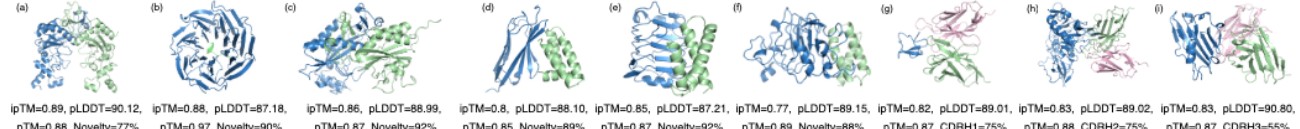

*Figure 3.* Designed protein complexes from: (a-c) general protein-protein complex design, (d-f) target-protein mini-binder complex design, and (g-i) antigen-antibody complex design. Target proteins are shown in blue, and the designed binder proteins in green, with light chains in pink for antigen-antibody complexes. PPDiff is able to design high-affinity protein-binding proteins across diverse target scaffolds.

### 5.4. Impact of Additional Swiss-Prot Data Pretraining

To assess whether pretraining on additional data improves design quality, we pretrain PPDiff on the Swiss-Prot dataset (Boeckmann et al., 2003) and its AlphaFold-derived structures (Varadi et al., 2022). Since Swiss-Prot proteins are monomers, pseudo complexes are generated by identifying residue pairs within 5Å in the 3D structure and with sequence length longer than 60, designating the interacting residues as the last and first residues of two chains. The model is first pretrained on this pseudo complex dataset before continuing training on PPBench. Figure 2 (c) and (d) compare the pLDDT and success rates of top-1, top-5, and top-10 candidates from this model with those of the original PPDiff. **The results indicate that pretraining on additional Swiss-Prot data provides no significant benefits.** Detailed results are included in the Appendix B.2.

### 5.5. Influence of Total Candidate Size

To evaluate the impact of total candidate size, we generate 30 binder proteins for each target protein in the test set of PPBench and analyze the top-1, top-5, and top-10 results, as shown in Figure 2 (f-h). Detailed results are provided in Appendix B.3. The results indicate that increasing the total candidate size to 100 significantly enhances the quality

of top-$k$ candidates ($k$=1,5, and 10), particularly in terms of success rate. This demonstrates that PPDiff can produce higher-quality complexes when considering a larger pool of candidates, which is potentially advantageous for wet-lab validation in real-world applications.

### 5.6. In-Depth Exploration of Target Protein–Mini Binder Complex Design

To better evaluate the quality of our designed binders, we first compute the docking scores between the target proteins and the designed binders using HDOCK (Yan et al., 2020). The results, summarized in Table 4, show that PPDiff achieves better docking performance to existing baseline methods, exhibiting similar phenomenon using AF2_pAE interaction scores. This validates both the reliability of the AF2 pAE_interaction score as a proxy for binder quality and the ability of PPDiff to generate binders with strong binding affinity to the target proteins.

We further compare PPDiff against a strong baseline, RFD-iffusion combined with ProteinMPNN (RF+MPNN), with results presented in Table 5. Our model achieves a higher average success rate across the 10 target proteins. Moreover, the binders generated by PPDiff exhibit significantly greater diversity and novelty compared to those produced

| Models | Seen Class | | | | | Zero-Shot | | | | | Average |
| --- | --- | --- | --- | --- | --- | --- | --- | --- | --- | --- | --- |
| | FGFR2 | InsulinR | PDGFR | TGFb | VirB8 | H3 | IL7Ra | EGFR | TrkA | Tie2 | |
| SEnc+ProteinMPNN | -197.82 | -192.56 | -231.46 | -203.41 | -235.67 | -198.23 | -192.85 | -178.23 | -224.91 | -201.32 | -205.64 |
| InterleavingDiff | -230.40 | -233.49 | -234.60 | -231.93 | -222.56 | -227.03 | -229.93 | -218.26 | -234.12 | -230.43 | -230.25 |
| SSINC Network | -207.76 | -193.18 | -226.77 | -211.04 | -220.34 | -206.02 | -207.24 | -183.53 | -217.91 | -196.22 | -208.48 |
| PPDiff | **-256.86** | **-260.95** | **-270.55** | **-251.35** | **-252.69** | **-244.23** | **-261.36** | **-244.75** | **-266.06** | **-265.19** | **-256.45** |

*Table 4.* The docking score ($\downarrow$) for all models on the target-protein mini-binder complex design task. "Average" refers to the overall docking score across the entire test set, rather than a simple mean across target categories. Our proposed PPDiff demonstrates the best binding affinity across all targets, outperforming all baseline methods.

| Models | Seen Class | | | | | Zero-Shot | | | | | Average | Nov. | Div. |
| --- | --- | --- | --- | --- | --- | --- | --- | --- | --- | --- | --- | --- | --- |
| | FGFR2 | InsulinR | PDGFR | TGFb | VirB8 | H3 | IL7Ra | EGFR | TrkA | Tie2 | | | |
| RF+MPNN | **28.07%** | 8.69% | **15.38%** | 22.22% | **57.14%** | 7.89% | 28.57% | **25.00%** | **100.00%** | 0.0 | 21.46% | 78.10% | 25.71% |
| PPDiff | 7.36% | **10.43%** | 14.61% | **35.56%** | 11.42% | **55.26%** | **60.00%** | 0.0 | 20.00% | **30.00%** | **23.16%** | **91.39%** | **91.79%** |

*Table 5.* The success rate (**%**, $\uparrow$) for PPDiff and a strong baseline RFDiffusion+ProteinMPNN (called RF+MPNN here due to the limited space) on the target-protein mini-binder complex design task. "Average" refers to the overall success rate across the entire test set, rather than a simple mean across target categories. Our PPDiff achieves a higher average success rate, as well as significantly higher novelty (Nov.) and diversity (Div.) scores.

by the baseline. These findings highlight the effectiveness of PPDiff in designing functional, diverse, and novel protein binders.

### 5.7. Designing Novel and Diverse Protein Complexes

Figure 3 showcases nine designed protein-protein complexes across different design tasks: (a-c) general protein-protein complex design, (d-f) target-protein mini-binder complex design, and (g-i) antigen-antibody complex design. The target proteins in these examples exhibit diverse structural scaffolds, and all complexes achieve outstanding metrics, including ipTM scores approaching 0.8 or higher, pTM scores exceeding 0.8, pLDDT values above 80, and novelty scores surpassing 50%. These results highlight **PPDiff's capability to design novel, high-affinity binder proteins for a wide range of protein targets**. More designed complexes are provided in Appendix D.

## 6. Conclusion

In this paper, we present PPDiff, a diffusion model building upon SSINC network to co-design binder protein sequences and structures for specified protein targets. SSINC incorporates interleaved self-attention layers and $k$NN-based equivariant graph layers, complemented by causal attention layers, enabling it to jointly model sequences, structures, and their complicated interdependencies. The model is pretrained on a general protein-protein complex design task and fine-tuned for two critical real-world applications, achieving success rates of 50.00%, 23.16% and 16.89%, respectively. A potential limitation of this work is the absence of wet-lab validation, which is an essential step planned for our future

research to further validate the model's practical utility.

## Impact Statement

This paper presents work whose goal is to advance the field of Machine Learning. There are many potential societal consequences of our work, none of which we feel must be specifically highlighted here.

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

# Appendix

## A. Data Statistics

### A.1. Data Statistics for PPBench

Detailed data statistics for the curated protein-protein complex dataset are reported in Table 6.

|  | Cluster | Complex |
|---|---|---|
| Training | 22,827 | 706,149 |
| Validation | 10 | 155 |
| Test | 10 | 56 |
| Total | 22,847 | 706,360 |

*Table 6.* Data statistics for PPBench.

| Models | Seen Class | | | | | Zero-Shot | | | | |
|---|---|---|---|---|---|---|---|---|---|---|
|  | FGFR2 | InsulinR | PDGFR | TGFb | VirB8 | H3 | IL7Ra | EGFR | TrkA | Tie2 |
| Training | 465 | 192 | 210 | 77 | 57 | – | – | – | – | – |
| Validation | 57 | 23 | 26 | 9 | 7 | – | – | – | – | – |
| Test | 57 | 23 | 26 | 9 | 7 | 38 | 7 | 4 | 4 | 2 |
| Total | 579 | 238 | 262 | 95 | 71 | 38 | 7 | 4 | 4 | 2 |

*Table 7.* Detailed data statistics for curated target-protein mini-binder complex dataset.

### A.2. Data Statistics for Target-Protein Mini-Binder Complex Dataset

We collect experimentally confirmed positive target-protein mini-binder complexes against ten target proteins with diverse structural properties from Bennett et al. (2023). The detailed data statistics for the ten target proteins are reported in Table 7.

|  | CDR-H1 Clustering | CDR-H2 Clustering | CDR-H3 Clustering |
|---|---|---|---|
| Complexes | 4,261 | 4,261 | 4261 |
| Clusters | 1,034 | 1,418 | 2,254 |
| Training | 3,590 | 3,576 | 3,427 |
| Validation | 403 | 399 | 451 |
| Test | 268 | 286 | 383 |

*Table 8.* Detailed data statistics for antigen-antibody complex dataset.

### A.3. Data Statistics for Antigen-Antibody Complex Dataset

The detailed data statistics for each CDR clustering are reported in Table 8.

## B. Supplementary Experimental Results

### B.1. Antigen-Antibody Complex Design Performance for Each CDR Clustering

Model performance for each CDR clustering is provided in Table 9, 10 and 11. Our proposed PPDiff achieves higher success rate than the strongest baseline SSINC Network on average and obtains much higher novelty scores.

### B.2. Swiss-Prot Data Pretraining

To evaluate if pretraining our PPDiff on additional data will improve model performance, we first pretrain our model on curated pseudo Swiss-Prot complexes, and then continue training the model on our constructed PPBench. We compare this

| Methods | ipTM (↑) | pTM (↑) | PAE (↓) | pLDDT(↑) | Success Rate (↑) | H1 Novelty (↑) | H2 Novelty (↑) | H3 Novelty (↑) |
|---|---|---|---|---|---|---|---|---|
| SEnc +ProteinMPNN | 0.364 | 0.566 | 15.021 | 76.063 | 4.85% | 60.37% | 60.95% | 69.39% |
| InterleavingDiff | 0.426 | 0.621 | 14.993 | 81.581 | 7.83% | **62.38%** | 68.21% | 77.48% |
| SSINC Network | **0.501** | 0.636 | **12.759** | **82.608** | **14.55%** | 34.87% | 38.50% | 57.55% |
| PPDiff | 0.484 | **0.638** | 14.058 | 82.509 | 11.11% | 61.62% | **69.69%** | **77.62%** |

*Table 9.* Model performance on antigen-antibody complex design task for CDR-H1 clustering. (↑): the higher the better. (↓): the lower the better. Novelty scores for CDR-H1, CDR-H2, and CDR-H3 are reported as H1, H2, and H3 novelty, respectively. PPDiff achieves the highest pTM and novelty scores across the designed CDR-H2 and CDR-H3.

| Methods | ipTM (↑) | pTM (↑) | PAE (↓) | pLDDT(↑) | Success Rate (↑) | H1 Novelty (↑) | H2 Novelty (↑) | H3 Novelty (↑) |
|---|---|---|---|---|---|---|---|---|
| SEnc +ProteinMPNN | 0.525 | 0.629 | 13.339 | 79.260 | 7.34% | **62.23%** | 59.71% | 74.66% |
| InterleavingDiff | 0.539 | 0.681 | 12.546 | 81.155 | 17.48% | 55.54% | 59.03% | 74.51% |
| SSINC Network | 0.547 | 0.658 | 11.929 | 82.829 | 20.27% | 30.29% | 39.94% | 61.13% |
| PPDiff | **0.561** | **0.682** | 12.535 | **83.162** | **22.03%** | 57.77% | **65.31%** | **75.95%** |

*Table 10.* Model performance on antigen-antibody complex design task for CDR-H2 clustering. (↑): the higher the better. (↓): the lower the better. Novelty scores for CDR-H1, CDR-H2, and CDR-H3 are reported as H1, H2, and H3 novelty, respectively. PPDiff achieves the highest scores on almost all metrics except CDR-H1 novelty score.

| Methods | ipTM (↑) | pTM (↑) | PAE (↓) | pLDDT(↑) | Success Rate (↑) | H1 Novelty (↑) | H2 Novelty (↑) | H3 Novelty (↑) |
|---|---|---|---|---|---|---|---|---|
| SEnc +ProteinMPNN | 0.457 | 0.539 | 14.894 | 78.036 | 6.52% | **62.75%** | 57.93% | 68.49% |
| InterleavingDiff | 0.523 | 0.659 | 13.210 | 80.082 | 10.70% | 54.69% | 63.53% | 74.75% |
| SSINC Network | 0.569 | 0.668 | **11.506** | 82.682 | 15.40% | 26.16% | 34.67% | 60.91% |
| PPDiff | **0.574** | **0.683** | 12.405 | **82.810** | **17.52%** | 53.98% | **64.18%** | **74.94%** |

*Table 11.* Model performance on antigen-antibody complex design task for CDR-H3 clustering. (↑): the higher the better. (↓): the lower the better. Novelty scores for CDR-H1, CDR-H2, and CDR-H3 are reported as H1, H2, and H3 novelty, respectively. PPDiff achieves the highest scores on the antigen-antibody binding affinity (ipTM) and success rate.

| | Methods | ipTM (↑) | pTM (↑) | PAE (↓) | pLDDT(↑) | Success Rate (↑) | Novelty (↑) | Diversity (↑) |
|---|---|---|---|---|---|---|---|---|
| | Ground Truth | 0.691 | 0.782 | 7.901 | 86.987 | 69.64% | – | – |
| top-1 | PPDiff | **0.700** | **0.779** | **9.153** | **83.765** | 50.00% | **89.46%** | – |
| | - With pretraining | 0.666 | 0.770 | 9.647 | 82.952 | **53.57%** | 88.89% | – |
| top-5 | PPDiff | **0.665** | **0.747** | 10.231 | **81.659** | **45.71%** | 88.93% | 90.76% |
| | - With pretraining | 0.663 | 0.733 | **10.087** | 79.597 | 35.36% | **89.90%** | **91.81%** |
| top-10 | PPDiff | **0.633** | **0.729** | **10.895** | **80.322** | **37.68%** | 89.10% | 91.09% |
| | - With pretraining | 0.626 | 0.710 | 11.255 | 78.068 | 27.68% | **90.74%** | **92.17%** |

*Table 12.* Model performance on additional Swiss-Prot data pretraining. (↑): the higher the better. (↓): the lower the better. Calculating the diversity score for the top-1 candidate is unnecessary, as it consists of only a single candidate.

model and our original PPDiff in Table 12. It shows pretraining on Swiss-Prot data does not provide significant benefit.

## B.3. Smaller Candidate Pool Size

To study the effect of total candidate size, we provide the results for a candidate pool size of 30 in Table 13. It shows reducing the candidate pool size leads to worse model performance on almost all metrics.

| | Methods | ipTM (↑) | pTM (↑) | PAE (↓) | pLDDT(↑) | Success Rate (↑) | Novelty (↑) | Diversity (↑) |
|---|---|---|---|---|---|---|---|---|
| | Ground Truth | 0.691 | 0.782 | 7.901 | 86.987 | 69.64% | – | – |
| top-1 | SEnc +ProteinMPNN | 0.586 | 0.693 | 12.029 | 77.064 | 28.57% | 63.28% | – |
| | InterleavingDiff | 0.631 | 0.722 | 11.199 | 79.651 | 33.92% | **91.61%** | – |
| | SSINC Network | 0.634 | 0.736 | **10.256** | 81.461 | 37.50% | 37.60% | – |
| | PPDiff | **0.644** | **0.739** | 10.289 | **81.926** | **39.28%** | 88.76% | – |
| top-5 | SEnc +ProteinMPNN | 0.569 | 0.683 | 12.931 | 76.201 | 14.28% | 65.37% | 62.21% |
| | InterleavingDiff | 0.594 | 0.695 | 12.594 | 76.465 | 15.00% | **91.69%** | **92.09%** |
| | SSINC Network | 0.602 | 0.709 | 11.528 | **80.800** | **32.14%** | 37.26% | 15.04% |
| | PPDiff | **0.606** | **0.713** | **11.459** | 79.176 | 30.36% | 89.11% | 91.45% |
| top-10 | SEnc +ProteinMPNN | 0.553 | 0.662 | 13.641 | 75.148 | 7.14% | 67.89% | 63.92% |
| | InterleavingDiff | 0.581 | 0.677 | 13.512 | 74.090 | 7.50% | **91.60%** | **92.49%** |
| | SSINC Network | 0.586 | 0.691 | **11.856** | **78.920** | **21.42%** | 37.43% | 15.18% |
| | PPDiff | **0.592** | **0.695** | 12.394 | 77.014 | 18.21% | 90.18% | 91.68% |

*Table 13.* Model performance with candidate pool size set to 30. (↑): the higher the better. (↓): the lower the better. Calculating the diversity score for the top-1 candidate is unnecessary, as it consists of only a single candidate.

## C. Additional Experimental Details

### C.1. PPBench Construction Details

We curate a large-scale protein-protein complex dataset by identifying chain-pair interfaces, following the data processing pipeline introduced in AlphaFold3 (Abramson et al., 2024). The curation process begins by applying a series of quality filters to identify valid PDB entries:

- Retain structures with a reported resolution of 9Å or better.

- Remove polymer chains containing unknown residues.

- Exclude protein chains where consecutive $C_\alpha$ atoms are separated by more than 10Å.

- Filter out polymer chains with fewer than four resolved residues.

- Eliminate clashing chains, defined as chains with more than 30% of their atoms located within 1.7Å of an atom in another chain. When two chains clashed:
  - The chain with the higher percentage of clashing atoms is removed.
  - If the clashing percentage is equal, the chain with fewer total atoms is removed.
  - If both chains have the same number of atoms, the chain with the larger chain ID is removed.

This filtering process result in 126,569 valid PDB entries. From these entries, chain-pair interfaces are identified, defined as pairs of chains with a minimum heavy atom separation of less than 5Å. This step yields a total of 367,016 chain-pair interfaces.

### C.2. Model Training Details

The embedding dimensionality is configured at 1,280. PPDiff is trained for 1,000,000 steps on a single NVIDIA RTX A6000 GPU using the Adam optimizer (Kingma, 2014). The batch size is set to 1,024 tokens, and the learning rate is initialized at 5e-6 . For structure diffusion, the starting and ending values of $\beta$ are set to 1e-7 and 2e-3, respectively, with a variance schedule of 2. The cosine schedule offset for sequence diffusion is configured at 0.01. The number of $k$-nearest neighbors is fixed at 32. Additionally, a learning rate warm-up is applied over the first 4,000 steps to stabilize the training process.

### C.3. Detailed Explanation of Metrics

We provide the detailed explanation of metrics as follows:

- **ipTM** is an interfacial variant of the predicted TM-score, evaluating the interaction between different chains (Abramson et al., 2024). Values higher than 0.8 represent confident high-quality predictions.

- **pTM** is a confidence score estimating the accuracy of global protein structure prediction. A pTM score above 0.5 means the overall predicted fold for the complex might be similar to the true structure (Zhang & Skolnick, 2004).

- **PAE** estimates the error in the relative position and orientation between two tokens in the predicted structure. Higher values indicate higher predicted error and therefore lower confidence.

- **pLDDT** aims to predict a modified LDDT score that only considers distances to polymers. It is a per-atom confidence estimate on a 0-100 scale where a higher value indicates higher structural stability.

### C.4. Target-Protein Mini-Binder Complex Design Evaluation Details

To calculate the pAE_interaction score, we first predict the structure of the designed binder sequence using ESMFold (Lin et al., 2023). The predicted structure is then superimposed onto the experimentally validated positive binder, and the AF2 pAE_interaction score is calculated for the resulting target-protein mini-binder complex. Using these scores, we determine the **success rate** by generating five mini-binder candidates for each target protein. The success rate is defined as the proportion of designed binders that achieve a lower pAE_interaction score with the target protein than the ground truth positive binder:

$$\text{SR} = \frac{1}{|\mathcal{D}_{\text{test}}| * k} \sum_{\mathcal{T} \in \mathcal{D}_{\text{test}}} \sum_{i=1}^{k} \text{II}(f(\mathcal{T}, \mathcal{B}_i, \mathcal{B})) \tag{16}$$

where $k=5$, $f(\mathcal{T}, \mathcal{B}_i, \mathcal{B}) = \text{true}$ if the $i$-th binder candidate $\mathcal{B}_i$ for target protein $\mathcal{T}$ achieves a lower pAE_interaction score than the positive binder $\mathcal{B}$. Otherwise, $f(\mathcal{T}, \mathcal{B}_i, \mathcal{B}) = \text{false}$.

## D. Additional Designed Cases

We provide more designed protein-protein complexes for general protein-protein complex design, target-protein mini-binder complex design and antigen-antibody complex design in Figure 4, 5 and 6, respectively. It shows our PPDiff is capable of designing novel and high-affinity protein-binding proteins for diverse protein targets across a wide range of domains.

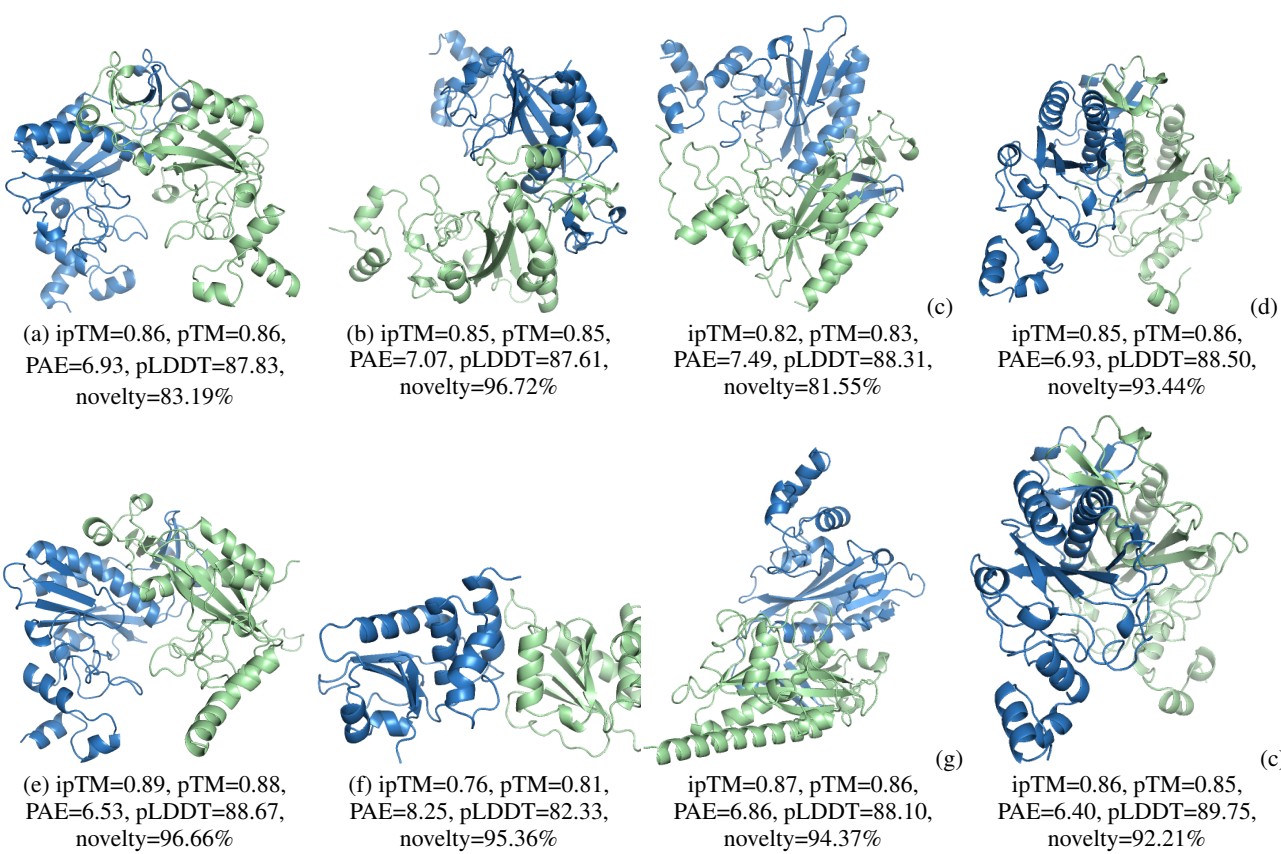

(a) ipTM=0.86, pTM=0.86, PAE=6.93, pLDDT=87.83, novelty=83.19%

(b) ipTM=0.85, pTM=0.85, PAE=7.07, pLDDT=87.61, novelty=96.72%

(c)
ipTM=0.82, pTM=0.83, PAE=7.49, pLDDT=88.31, novelty=81.55%

(d)
ipTM=0.85, pTM=0.86, PAE=6.93, pLDDT=88.50, novelty=93.44%

(e) ipTM=0.89, pTM=0.88, PAE=6.53, pLDDT=88.67, novelty=96.66%

(f) ipTM=0.76, pTM=0.81, PAE=8.25, pLDDT=82.33, novelty=95.36%

(g)
ipTM=0.87, pTM=0.86, PAE=6.86, pLDDT=88.10, novelty=94.37%

(c)
ipTM=0.86, pTM=0.85, PAE=6.40, pLDDT=89.75, novelty=92.21%

*Figure 4.* Designed complexes for general protein-protein complex design by our PPDiff. All of them achieve an ipTM score approaching or higher than 0.8, pTM score above 0.8, PAE lower than 10 and pLDDT better than 80. These designed binder sequences also have novelty scores higher than 80%, validating that PPDiff is capable of designing novel and high-affinity protein-binding proteins across diverse protein targets.

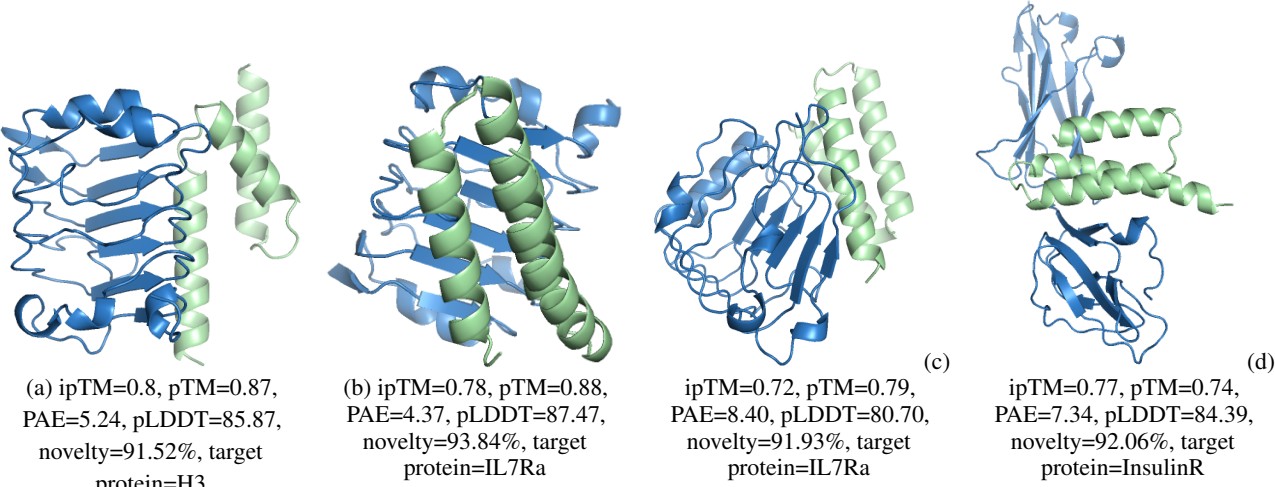

(a) ipTM=0.8, pTM=0.87, PAE=5.24, pLDDT=85.87, novelty=91.52%, target protein=H3

(b) ipTM=0.78, pTM=0.88, PAE=4.37, pLDDT=87.47, novelty=93.84%, target protein=IL7Ra

(c) ipTM=0.72, pTM=0.79, PAE=8.40, pLDDT=80.70, novelty=91.93%, target protein=IL7Ra

(d) ipTM=0.77, pTM=0.74, PAE=7.34, pLDDT=84.39, novelty=92.06%, target protein=InsulinR

*Figure 5.* Designed complexes by our PPDiff for target-protein mini-binder complex design. All of them achieve an ipTM score higher than 0.7, pTM score above 0.7, PAE lower than 10 and pLDDT better than 80. These designed binder sequences also have novelty scores higher than 80%, validating that PPDiff is capable of designing novel and high-affinity binder proteins across diverse protein targets.

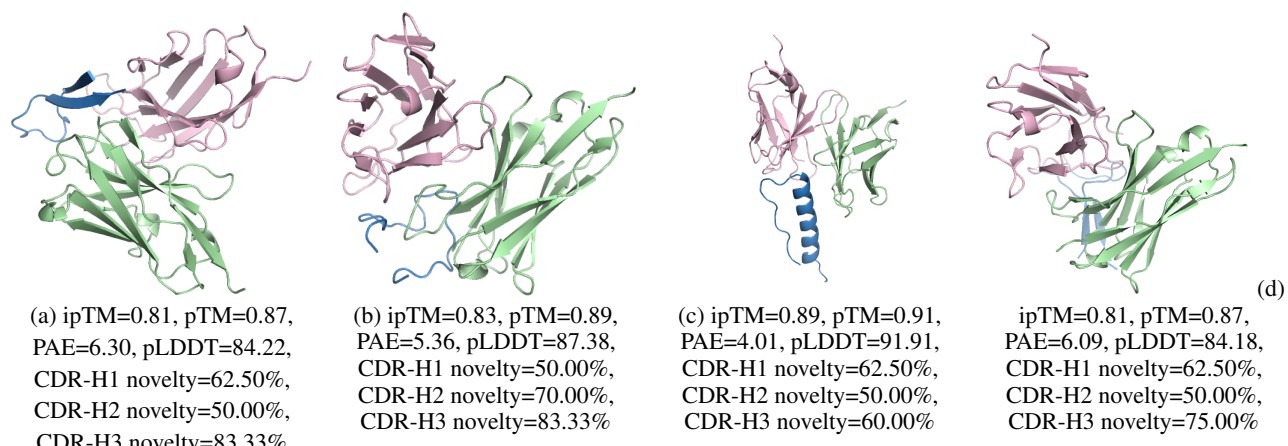

(a) ipTM=0.81, pTM=0.87, PAE=6.30, pLDDT=84.22, CDR-H1 novelty=62.50%, CDR-H2 novelty=50.00%, CDR-H3 novelty=83.33%

(b) ipTM=0.83, pTM=0.89, PAE=5.36, pLDDT=87.38, CDR-H1 novelty=50.00%, CDR-H2 novelty=70.00%, CDR-H3 novelty=83.33%

(c) ipTM=0.89, pTM=0.91, PAE=4.01, pLDDT=91.91, CDR-H1 novelty=62.50%, CDR-H2 novelty=50.00%, CDR-H3 novelty=60.00%

(d) ipTM=0.81, pTM=0.87, PAE=6.09, pLDDT=84.18, CDR-H1 novelty=62.50%, CDR-H2 novelty=50.00%, CDR-H3 novelty=75.00%

*Figure 6.* Designed complexes by our PPDiff for antigen-antibody complex design. All of them achieve an ipTM score higher than 0.8, pTM score above 0.8, PAE lower than 10 and pLDDT better than 80. These designed binder sequences also have CDR-H1, H2 and H3 novelty scores higher than 50%, validating that PPDiff is capable of designing novel and high-affinity antibodies for antigens.

