# OpenReview forum: "PPDiff: Diffusing in Hybrid Sequence-Structure Space for Protein-Protein Complex Design"
_ICML.cc/2025/Conference — ICML 2025 poster_

### Official Review · Reviewer_c1qo · 2025-03-11

**Overall Recommendation:** 3

**Summary:**

The paper focuses on the task of generative binder design, modeling protein-protein complexes. This is a critical and impactful task in protein design. The authors introduce *PPDiff*, which uses a protein sequence and backbone co-design strategy during generation leveraging a joint diffusion framework. An important component of the work is the new neural network architecture, which interleaves self-attention layers for sequence processing with equivariant graph layers to capture structure details. Furthermore, the work curates a new dataset, PPBench, consisting of around 700k protein complexes for training. The model is extensively validated on several binder design tasks and demonstrates strong success rates. The authors also extensively ablate and analyze their approach and the design choices.

**Claims And Evidence:**

The paper's main claim is the development of a novel binder design generation model with high success rates. This claim is extensively validated in numerical experiments. The presented experimental results are indeed promising and evidence for PPDiff's strong binder generation performance. There are no concerns regarding claims and evidence.

**Essential References Not Discussed:**

While overall the discussion of related work is appropriate, I believe some key citations are missing. The work relies on a joint sequence-structure co-generation framework. Maybe the first work to do such co-generation for protein design was *MultiFlow* (https://arxiv.org/abs/2402.04997, ICML 2024). This work is not cited or discussed. Moreover, the generation process of the categorical residue identities seems to exactly correspond to the process first described in the seminal *D3PM* paper, (https://arxiv.org/abs/2107.03006, NeurIPS 2021). Also this work is not cited or discussed. The authors only cite Guan et al., but I believe these two papers are even more important.

Finally, a recent influential work for generative binder design is *BindCraft* (https://www.biorxiv.org/content/10.1101/2024.09.30.615802v1). This work also is not mentioned in the paper. I think this is okay, because BindCraft can be considered concurrent, and therefore this does not affect my paper rating. However, it would certainly make the paper stronger if also BindCraft was discussed and ideally evaluated as an additional baseline -- this can be considered an optional suggestion to the authors.

**Experimental Designs Or Analyses:**

Yes, I checked the soundness and validity of the experimental designs and analyses, but did not identify and major issues.

**Methods And Evaluation Criteria:**

The proposed methods and evaluation criteria do overall make sense and are appropriate for the problem at hand. I do have some concerns, though:

1. PPDiff jointly generates sequences and structures. All quantitative evaluations, however, seem to be carried out on the structures based on the folded sequences with AF3 or AF2. This means the structure that is co-generated from PPDiff itself is not actually scored at all or used in any evaluations, if I understand the work correctly. An obvious question is, what is the co-designability score? Does the generated sequence actually fold into the structure that is output by PPDiff itself (e.g. what is the RMSD of the generated structure against the folded structure based on the sequence)? I believe this should be analyzed and evaluated for any model that jointly generates sequences and structures. This corresponds to the co-designability metric that is, for instance, used here, "Generative Flows on Discrete State-Spaces: Enabling Multimodal Flows with Applications to Protein Co-Design", https://arxiv.org/abs/2402.04997.

2. As one baseline the authors choose a purely structure-based generator together with ProteinMPNN ("SEnc+ProteinMPNN"), and the authors point out that there are generally no baseline models to compare to. While it is true that there are no "standard" binder design benchmarks, I do believe that as an additional baseline, similar to the mentioned one, it would be appropriate to run RFDiffusion with ProteinMPNN, as it is known that RFDiffusion has been successfully used for binder design. Quite possibly, RFDiffusion+ProteinMPNN would perform better than the SEnc+ProteinMPNN baseline, and this would be the only true "not-self-designed" baseline.

It would be great if the authors could comment on these issues.

**Other Comments Or Suggestions:**

I strongly encourage the authors to publicly release the PPBench dataset, as well as the separate curated datasets for the mini-binder and antibody/antigen design experiments.

**Other Strengths And Weaknesses:**

**Strengths:**
- The paper is generally well written and (mostly) easy to follow.
- I appreciate the paper's data engineering and the curation of the new PPBench dataset.
- The extensive analyses and ablation experiments are insightful.
- The paper demonstrates strong binder generation performance, according to the success metrics, an important task that has not been studied that broadly in the machine learning literature. This makes the work significant.
- The paper provides an anonymous link to source code. I did not check this code, but I appreciate the sharing of the code.

**Weaknesses:**
- There are some concerns regarding the method's evaluation (see above).
- There are some concerns regarding the discussion of related work (see above).
- Some aspects in the paper are not well-explained and some details seem slightly incorrect (see "Question For Authors" below).
- The paper makes some choices when designing the method that are not clear or well-motivated, see "Question For Authors" below.

**Questions For Authors:**

1. In Section 3.4, the generation process of the binder protein is described in detail. However, it is not clear how exactly the target protein is fed to the model as conditioning, this is, how exactly the target sequence/structure enter the self-attention or equivariant graph layers. Also in figure 1, this is not clear. Can the authors clarify this? The conditioning implementation can be critical for strong performance, but this is not clear.

2. I believe equation (7) is not correct. The way the generation process is written in the first line of equation (7) means that there is a strict independence between the sequence and structure generation. However, this is not true, but sequence and structure are generated jointly, dependent on each other with one network processing both. This is, $s_{t-1}$ depends both on $s_t$ and $x_t$, and similarly $x_{t-1}$ also depends on both. I think we should have $... = p_\theta(s^B_{t-1} |s^B_t, x^B_t, T) p_\theta(x^B_{t-1} |x^B_t, s^B_t, T)$. This applies to everywhere in the paper, where any of the $p_\theta$ occur. I would like the authors to comment on that or clarify, if I am misunderstanding something.

3. The authors add a causal attention layer to the model, which improves performance. However, why does this layer need to be *causal*? This is not well-motivated or discussed. Why not a regular attention layer instead? This causal layer imposes a direction in the sequence, but there is no natural direction in the protein sequence.

4. "As suggested by Ho et al. (2020), we set $\lambda_t=1$, ...". Ho et al. suggested this for the epsilon/noise prediction setting, not for $x_0$ prediction, so this sentence does not seem appropriate.

5. In Eq. (13), the authors propose a more informative prior. That such a design choice in the method is relevant should be supported by an ablation experiment training with and without this informative prior, but such an experiment is missing.

6. Also in Section 3.6, the authors write *"For sequence guidance, we randomly sample secondary structure fragments
from the training dataset, identified by using DSSP. ..."* The authors should provide details what exactly they are doing here and also run an ablation over this design choice. This is a rather unusual choice, as most discrete diffusion models start their generation process sampling from a uniform categorical distribution (or all masked), but not with samples from the dataset. This may also bias the generation. I would like the authors to comment on this.

7. Why exactly do the mini-binder generation experiments use the AF2 pAE_interaction score for evaluation and the other experiments the AF3-based ipTM, pTM, PAE and pLDDT scores? Wouldn't it be better to use all scores in all experiments?

8. The authors initialize their self-attention layers from ESM2. How important is this? How would the model perform if those layers were initialized randomly? An ablation experiments over this would be quite relevant, too.

**Relation To Broader Scientific Literature:**

Overall, the work is appropriately positioned with respect to the broader scientific literature. The paper's introduction is very informative in that regard and discusses prior approaches. A dedicated related work section extends the discussion.

**Theoretical Claims:**

The paper does not have any advanced mathematical proofs or theoretical claims. The mathematical details in the method section seem mostly correct. There are some issues (discussed later), but these do not represent major flaws.

---

> ### Author Rebuttal · Authors · 2025-03-31
>
> We appreciate the reviewer's dedication in providing detailed feedback. We have clarified all the reviewer’s concerns and conducted additional experiments accordingly. All ablation studies were conducted by generating one candidate and we evaluated the resulting complexes using AF3 with the same seed. Responses to specific points are provided below:
>
> **Q1: What is the co-designability score?**
>
> Ans: We evaluated three consistency metrics for the top-1candidate: (1) Seq RMSD: RMSD between the folded structure of the designed sequence and ground truth, (2) Struct RMSD: RMSD between the designed structure and ground truth, (3) Design scRMSD: RMSD between the folded structure of the generated sequence and the designed structure. Although Struct RMSD and Design scRMSD are higher than Seq RMSD, incorporating structural information significantly enhances binder sequence design. As we can see, PPDiff achieves a substantially lower Seq RMSD (1.13) than a sequence-only model (5.79, Finetuned ESM2 on PPBench), validating the importance of co-design in producing sequences that closely align with their intended structures.
>
> ||Seq RMSD|Struct RMSD|Design scRMSD|
> |--|--|--|--|
> |Finetuned ESM2|5.79|--|--|
> |SEnc +ProteinMPNN|6.24|**6.03**|7.46|
> |InterleavingDiff| 2.89|6.45|7.51|
> |SSINC Network|1.23|6.89|7.94|
> |PPDiff |**1.13**|6.32|**6.87**|
>
> **Q2: It would be appropriate to run RFDiffusion with ProteinMPNN.**
>
> Ans:  Please refer to the **response of reviewer zYHd Q1.**
>
> **Q3: Essential References Not Discussed**
>
> Ans: We appreciate the reviewer's valuable suggestions. We will cite and discuss MultiFlow, D3PM, and BindCraft in the revised version. We kindly note that our discrete sequence diffusion part is based on [1] as introduced in Sec. 3.3 line 144-148 in our manuscript, which precedes and is referenced by D3PM.
>
> [1] Argmax flows and multinomial diffusion: Learning categorical distributions. Hoogeboom et al. NeurIPS 2021.
>
> **Q4: About dataset release**
>
> Ans: We will release all datasets shortly.
>
> **Q5: Can the authors clarify how the target protein is fed to the model as conditioning?**
>
> Ans: We concatenate the sequences of the target and binder proteins, and represent all residues in target and binder proteins jointly as a unified point set.
>
> **Q6: In Eq (7), $s_{t-1}$  should depend on $s_t$ and $x_t$, and similarly $x_{t-1}$.**
>
> Ans: We appreciate the reviewer's suggestion and agree with the correction. We will update our manuscript in the revised version.
>
> **Q7:Why does the layer need to be causal?**
>
> Ans: Without causal attention layers, sequences displayed repetitive residue types (e.g., "EEEE"), resembling multimodality issues observed in non-autoregressive MT [2]. We introduced causal attention layers for autoregressive dependency management, significantly improving performance, as shown in our ablation study below.
>
> [2] Non-Autoregressive Neural Machine Translation. ICLR 2018.
>
> ||ipTM|pTM|PAE|pLDDT|
> |--|--|--|--|--|
> |PPDiff - Self Attention|0.562|0.642|15.153|70.185|
> |PPDiff - Casual Attention|**0.575**|**0.650**|**14.719**|**71.022**|
>
> **Q8: Ho et al. suggested $\lambda_t=1$ for the noise prediction setting, not for $x_0$ prediction**
>
> Ans: We apologize for any confusion. We set $\lambda_t=1$ following previous work [3]. We will accordingly update our manuscript.
>
> [3] 3D Equivariant Diffusion for Target-Aware Molecule Generation and Affinity Prediction. ICLR 2023.
>
> **Q9: The authors propose a more informative prior and sequence guidance, which should be supported by ablation studies.**
>
> Ans: Ablation studies, as presented below, clearly indicate that removing either the informative prior or sequence guidance decreases performance.
>
> | |ipTM|pTM|PAE|pLDDT|
> |--|--|--|--|--|
> |PPDiff|**0.575**|**0.650**|**14.719**|**71.022**|
> |- w/o informative prior|0.476|0.615|16.478|66.494|
> |- w/o sequence guidance|0.524|0.638|15.372|70.018|
>
> **Q10: Why exactly do the mini-binder generation experiments use the AF2 pAE_interaction score for binder evaluation**
>
> Ans: Following established practices in previous studies [4], we utilized the AF2 pAE_interaction score to evaluate binder designs due to its proven effectiveness in distinguishing experimentally validated binders from non-binders. Previous research has shown that binder candidates selected using AF2 pAE_interaction scores yield experimental success rates ranging from 1.5% to 7% across various target proteins [4].
>
> [4] Improving de novo protein binder design with deep learning. Nature Communication. 2023.
>
> **Q11: How important is the initialization of self-attention layers from ESM2?**
>
> Ans: We conducted an ablation study on removing ESM2 initialization below. Results indicate performance degradation upon removal, demonstrating the significance of leveraging ESM2's pretrained knowledge as an effective initialization for PPDiff.
>
> | |ipTM|pTM|PAE|pLDDT|
> |--|--|--|--|--|
> |PPDiff|**0.575**|**0.650**|**14.719**|**71.022**|
> |-w/o initialization|0.461|0.534|18.174|63.853|

---

### Official Review · Reviewer_RZ25 · 2025-03-12

**Overall Recommendation:** 4

**Summary:**

This paper presents PPDiff, a novel diffusion-based model for protein-protein complex design. The model aims to generate protein binders with high affinity for arbitrary target proteins by simultaneously designing both the sequence and structure of the binder. PPDiff builds upon the Sequence Structure Interleaving Network with Causal attention layers (SSINC). The authors introduce PPBench, a dataset consisting of 706,360 protein-protein complexes curated from the Protein Data Bank (PDB). The model is pretrained on PPBench and further fine-tuned on two key applications: Target-protein mini-binder complex design and Antigen-antibody complex design.

**Claims And Evidence:**

Yes.

**Essential References Not Discussed:**

N/A

**Experimental Designs Or Analyses:**

Yes. This work mainly used top-k success rate to represent the effectiveness of the model.

**Methods And Evaluation Criteria:**

Yes.

**Other Comments Or Suggestions:**

N/A

**Other Strengths And Weaknesses:**

This paper is well-written and presents a strong contribution. The authors propose a co-design framework for generating binders conditioned on a target, demonstrating superior performance compared to existing models. Unlike previous works, such as this study (https://openreview.net/pdf?id=dq3g7Bl9of), which focus more on backbone and sequence optimization, this paper emphasizes binder design, potentially making it applicable to a broader range of scenarios. Additionally, the experiments are comprehensive, with a thorough analysis of the impact of different components of the model. Regarding weaknesses, a common challenge in protein design models is the lack of experimental validation through wet-lab experiments. To strengthen the in silico validation, could the authors provide docking scores for the designed binders, particularly for well-known targets? This would help assess the binding efficacy and further support the model’s effectiveness.

**Questions For Authors:**

See above.

**Relation To Broader Scientific Literature:**

N/A

**Theoretical Claims:**

Yes. The formulation part of this paper clearly showed how to jointly diffuse protein sequence and structure.

---

> ### Author Rebuttal · Authors · 2025-03-31
>
> We greatly appreciate the reviewer for the insightful and constructive feedback. We have conducted additional experiments as suggested. Detailed responses to specific questions are provided below:
>
> **Q1: To strengthen the in silico validation, could the authors provide docking scores for the designed binders, particularly for well-known targets?**
>
> Ans: Thank you for this valuable suggestion. Because our PPDiff designs provide only alpha-carbon backbones, they are not directly applicable for docking. Therefore, we first used ESMFold to generate the structures of our designed binder sequences. We then performed docking simulations using HDOCK [1], pairing each predicted binder with its corresponding target protein. For each target, we produced five binder candidates and reported the average docking score for the top-1 candidate in Table 1. As noted in the original HDOCK paper, **a more negative docking score means a more possible binding model**; notably, this score should not be treated as the absolute measures of binding affinity because it has not been calibrated to the experimental data. The results show that **PPDiff consistently achieves more negative docking scores across all ten target proteins compared to baseline methods, indicating that binders designed by our model exhibit stronger potential affinities**. Additionally, Table 2 compares these scores with ground truth docking values, demonstrating that **PPDiff even outperforms experimentally confirmed binders (ground truth) in five categories** — a finding that highlights its potential for designing high-affinity protein binders.
>
> [1] The HDOCK server for integrated protein–protein docking. Yan et al. Nature Protocols. 2020.
>
> **Table1: Docking Scores Comparing with Baselines**
>
> |     | Seen Class |     |   |    |   | Zero-Shot |     |   |   |    | Average  |
> |------------|------|----|----|----|----|----|----|----|----|----|----------|
> |Target Protein|FGFR2|InsulinR|PDGFR|TGFb|VirB8|H3|IL7Ra|EGFR|TrkA|Tie2|Average|
> | SEnc +ProteinMPNN | -197.82 | -192.56  | -231.46 | -203.41 | -235.67 | -198.23   | -192.85 | -178.23 | -224.91 | -201.32 | -205.64  |
> | InterleavingDiff  | -230.40 | -233.49  | -234.60 | -231.93 | -222.56 | -227.03   | -229.93 | -218.26 | -234.12 | -230.43 | -230.25  |
> | SSINC Network     | -207.76 | -193.18  | -226.77 | -211.04 | -220.34 | -206.02   | -207.24 | -183.53 | -217.91 | -196.22 | -208.48  |
> | PPDiff  | **-256.86** | **-260.95**| **-270.55** | **-251.35** | **-252.69** | **-244.23**   | **-261.36** | **-244.75** | **-266.06** | **-265.19** | **-256.45**  |
>
> **Table2: Docking Scores Comparing with Ground Truth**
>
> |                | Seen Class |          |         |         |         | Zero-Shot |         |         |         |         | Average  |
> |----------------|------------|----------|---------|---------|---------|-----------|---------|---------|---------|---------|----------|
> | Target Protein | FGFR2      | InsulinR | PDGFR   | TGFb    | VirB8   | H3        | IL7Ra   | EGFR    | TrkA    | Tie2    | Average  |
> | Ground Truth   | -250.35 | **-339.78** | -218.2 | **-289.42** | **-282.15** | **-287.48** | -244.68 |**-316.38**| -227.56 | -261.83 | **-271.783** |
> | PPDiff         | **-256.86**    | -260.95  | **-270.55** | -251.35 | -252.69 | -244.23   | **-261.36** | -244.75 | **-266.06** | **-265.19** | -256.45  |

---

> > ### Comment · Reviewer_RZ25 · 2025-04-05
> >
> > The authors have addressed my concerns, and the docking score is generally better than the baselines. Therefore, I am revising my score.

---

> > > ### Author Response · Authors · 2025-04-05
> > >
> > > Thanks for taking the time to read our responses. We’re glad to hear that our responses are helpful, and we sincerely appreciate your updated score and constructive comments throughout the review process.

---

### Official Review · Reviewer_zYHd · 2025-03-13

**Overall Recommendation:** 3

**Summary:**

The authors propsoe a new diffusion method to tackle the protein complex problem. They define a co-design diffusion method that generates both the structure and the sequence of a protein complex. Then, they introduce a new architecture based on causal attention mechanism as well as knn equivariant layers. They pretrain this new model on a protein complex dataset issued from PDB (and SwissProt) and finetuned the model on 2 downstream tasks (antibody-antigen generation and mini-binder complex) and show state-of-the-art results.

**Claims And Evidence:**

They claim to define a new co-design method and this claim is true. They also claim that their method is competitive and this also seems true.

**Essential References Not Discussed:**

I think [1] should be discussed. It is a diffusion method based on framed

[1] Proteus: pioneering protein structure generation for enhanced designability and efficiency

**Experimental Designs Or Analyses:**

The experiments make sense and the analysis of their method is complete with a lot of ablation and sensitivity analysis (number of steps, different datasets, ...).

**Methods And Evaluation Criteria:**

The method makes a lot of sense and I like the perspective to challenge existing architecture. To the best of my knowledge, the evaluation makes sense and follows the standard procedure.

**Other Comments Or Suggestions:**

The authors did not cite framediff but framflow (page 2)

**Other Strengths And Weaknesses:**

The paper is well written.

**Questions For Authors:**

Can you finetune your method for the binder task and evaluate it against RFDiffusion? This would make the paper much stronger in my opinion.

**Relation To Broader Scientific Literature:**

This seems correct to me especially as most backbone generation method are trained for monomers and not protein complex generation.

**Theoretical Claims:**

Not applicable

---

> ### Author Rebuttal · Authors · 2025-03-31
>
> We sincerely thank the reviewer for the valuable comments and insightful suggestions. We have addressed all raised concerns and conducted additional experiments as recommended. Please find detailed responses to the specific points below:
>
> **W1:Essential References Not Discussed:I think [1] should be discussed. It is a diffusion method based on framed.**
>
> Ans: We appreciate the reviewer for highlighting this relevant reference. Proteus is an innovative frame-based diffusion approach, integrating a graph-based triangle technique and a multi-track interaction network, which shows robust capabilities in designing protein backbone structures. We will include a citation and thorough discussion of this method ([1]) in the revised manuscript.
>
> [1] Proteus: pioneering protein structure generation for enhanced designability and efficiency.
>
> **Comment: The authors did not cite framediff but framflow**
>
> Ans: Thank you for pointing out this oversight. We will appropriately cite the correct reference, Framediff ([2]), in our revised manuscript.
>
> [2] SE(3) diffusion model with application to protein backbone generation. Yim et al. ICML 2023.
>
> **Q1: Can you finetune your method for the binder task and evaluate it against RFDiffusion? This would make the paper much stronger in my opinion.**
>
> Ans: Thank you for this valuable suggestion. We initially followed the binder design procedure outlined in the RFDiffusion paper to generate the backbone structures. Subsequently, ProteinMPNN was employed to design corresponding binder sequences based on these backbone structures. To ensure a fair comparison, we pretrained ProteinMPNN on our curated PPBench and then fine-tuned the pretrained model specifically for the downstream binder design task. As RFDiffusion does not provide a training script, we utilized their publicly available model weights directly. Evaluation of success rates was conducted consistent with the methodology described in our manuscript. The average success rates across each target protein category are presented below. **Our results indicate that our model outperforms RFDiffusion + ProteinMPNN for 5 out of 10 target proteins, and achieves a higher overall average success rate across all tested target proteins. Notably, our PPDiff achieves much higher novelty and diversity scores, demonstrating our model's superior capability in designing high-affinity, novel and diverse binders**.
>
> **Table: Success Rate, Novelty and Diversity on Target Protein-Mini Binder Design Task**
>
> |                         | Seen Class |          |        |        |        | Zero-Shot |        |      |         |        | Average Success Rate | Novelty | Diversity |
> |-------------------------|------------|----------|--------|--------|--------|-----------|--------|------|---------|--------|---------|---------|-----------|
> | Target Protein          | FGFR2      | InsulinR | PDGFR  | TGFb   | VirB8  | H3        | IL7Ra  | EGFR | TrkA    | Tie2   | Average Success Rate | Novelty | Diversity |
> | RFDiffusion+ProteinMPNN | **28.07%**     | 8.69%    | **15.38%** | 22.22% | **57.14%** | 7.89%     | 28.57% | **25.00%**  | **100.00%** | 0.0    | 21.46%  | 78.10%  | 25.71%    |
> | PPDiff                  | 7.36%      | **10.43%**   | 14.61% | **35.56%** | 11.42% | **55.26%**    | **60.00%** | 0.0  | 30.00%  | **30.00%** | **23.16%**  | **91.39%**  | **91.79%**    |

---

### Official Review · Reviewer_oJL7 · 2025-04-02

**Overall Recommendation:** 2

**Summary:**

The paper presents a diffusion based generative model to create binding molecules for given protein targets. To do so the paper proposes a joint model which combines both the coordinates and types of residue sites. The major novelty resides in the score function network which alternates between self-attention and graph convolution layers. They then experiment with their model on a subset of PDB which they call PPBench. This seems to be another contribution in terms of dataset. On this data, the model shows improved accuracy and novelty.

**Claims And Evidence:**

I have not had the time to review this. I am placing some summary comments for aiding the review process, but do not expect any specific response for my concerns.

**Essential References Not Discussed:**

I have not had the time to review this. I am placing some summary comments for aiding the review process, but do not expect any specific response for my concerns.

**Experimental Designs Or Analyses:**

I have not had the time to review this. I am placing some summary comments for aiding the review process, but do not expect any specific response for my concerns.

**Methods And Evaluation Criteria:**

Diffusion model for protein binding challenges have been proposed before. Look at DiffBP: Generative Diffusion of 3D Molecules for Target Protein Binding and following works. None of these have been cited or compared as a baseline; especially as some of these are similar in flavor. For example diffbp while looking at molecules, almost straightforwardly can be used for protein protein binding as well.

**Other Comments Or Suggestions:**

I have not had the time to review this. I am placing some summary comments for aiding the review process, but do not expect any specific response for my concerns.

**Other Strengths And Weaknesses:**

I have not had the time to review this. I am placing some summary comments for aiding the review process, but do not expect any specific response for my concerns.

**Questions For Authors:**

The authors mention that they have their proposed network SSINC also as a baseline but without diffusion loss. I am confused as to how it generates the output then. Is it trained like an auto-regressive language model? However in the introduction they say that their model is non-autoregressive.

**Relation To Broader Scientific Literature:**

I have not had the time to review this. I am placing some summary comments for aiding the review process, but do not expect any specific response for my concerns.

**Theoretical Claims:**

I have not had the time to review this. I am placing some summary comments for aiding the review process, but do not expect any specific response for my concerns.

---

### Decision · Program_Chairs · 2025-05-01

**Decision:**

Accept (poster)

**Comment:**

The paper introduces a diffusion-based generative model for protein-protein complex design, which jointly generates sequences and structures using a co-design approach. The authors propose a novel neural architecture combining self-attention with equivariant graph layers and curate a large-scale dataset, PPBench, to support training and evaluation.

Reviewers highlight several strengths: strong performance on binder design tasks, clear evidence of co-design benefits, and competitive results in docking simulations. The rebuttal provides additional validation, including success rates, co-designability metrics, and docking scores compared to RFDiffusion and experimental binders. These responses address the main technical and evaluation concerns.

The paper contributes a new method, dataset, and thorough analysis. While some aspects (e.g., conditioning mechanism, evaluation setup) could be clearer and more standard comparisons (e.g., with BindCraft) would strengthen the work, the current version meets the bar for acceptance.